

# Polycyclic aromatic hydrocarbons (PAHs) in aerosols over the central Himalayas along two south-north transects

Peng Fei Chen[1,5], Chao Liu Li[1], Shi Chang Kang[2,3*], Maheswar Rupakheti[4], Arnico K Panday[6], Fang Ping Yan[2,5], Quan Lian Li[2], Qiang Gong Zhang[1,3], Jun Ming Guo[1,5], Dipesh Rupakheti[1,5], Wei Luo[7]

[1]Key Laboratory of Tibetan Environment Changes and Land Surface Processes, Institute of Tibetan Plateau Research, Chinese Academy of Sciences, Beijing 100101, China

[2]State Key Laboratory of Cryospheric Science, Cold and Arid Regions Environmental and Engineering Research Institute, Lanzhou 730000, China

[3]Center for Excellence in Tibetan Plateau Earth Sciences, Chinese Academy of Sciences, Beijing 100085, China;

[4]Institute for Advanced Sustainability Studies, Potsdam 14467, Germany

[5]University of Chinese Academy of Sciences, Beijing 100039, China

[6]International Centre for Integrated Mountain Development, Kathmandu, Nepal

[7]State Key Lab of Urban and Regional Ecology, Research Center for Eco-Environmental Sciences, Chinese Academy of Sciences, Beijing 100085, China



Correspondence to: Shi Chang Kang (shichang.kang@lzb.ac.cn)

**Abstract.** Our understanding of the transport of polycyclic aromatic hydrocarbons (PAHs) from the Indo–Gangetic Plains (IGP) to the Himalayas remains limited. Concentrations of PAHs were therefore measured in total suspended particles (TSP) from six sites along two south–north transects across the central Himalayas. Spatially, the annual average TSP and PAH (especially 5- and 6-ring) concentrations were found to decrease noticeably along both transects. The concentration levels of TSP and PAHs at Lumbini were found to be the highest (TSP: $209 \pm 113$ μg/m$^3$; PAHs: $91.6 \pm 54.6$ ng/m$^3$) which are comparable to those in some South Asian cities, but three and thirteen times higher than those at Nyalam (TSP: $59.1 \pm 62.0$ μg/m$^3$; PAHs: $5.57 \pm 3.36$ ng/m$^3$), respectively. The dry deposition fluxes also had a decreasing trend pattern from the southern to northern side of the Himalayas. Moreover, annual TSP and PAH concentrations exhibited a logarithmic decreasing pattern with increasing elevation especially in the non-monsoon seasons (TSP: y=-57.3lnx+552, $R^2$=0.952; PAHs: y=-26.8lnx+229, $R^2$=0.948). The TSP and PAH concentrations showed a clear seasonal variation, with the minimum concentrations (TSP: 47.9 μg/m$^3$; PAHs: 16.8 ng/m$^3$) around the mid-monsoon season and the maximum concentrations (TSP: 442 μg/m$^3$; PAHs: 192 ng/m$^3$) in the winter season at Lumbini. While at the remote sites (e.g. Nyalam and Zhongba), these pollutants were relatively constant throughout the year with relatively higher abundance during the pre-monsoon season. For example, approximately 80% of samples have PAH concentrations lower than 10 ng/m$^3$ at Nyalam and Zhongba. And just a few samples with higher PAH concentrations, however, not more than 21 ng/m$^3$, were observed during the pre-monsoon season. Both IndP/(IndP+BghiP) and Fla/(Fla+Pyr) ratios suggested that atmospheric PAHs from the Nepal sites were mainly associated with emission of biomass, coal burning and petroleum combustion. A similar composition pattern with relatively uniform contributions of 4 groups to total PAHs was found between the two sides of the Himalayas (e.g. Jomsom, Zhongba, and Nyalam), suggesting that the northern side of the Himalayas may be affected by anthropogenic emissions from the



IGP due to long-range transportation as well as the unique mountain/valley breeze system which bring pollution from the IGP into Tibet across the high Himalayas.

## 1    Introduction

Atmospheric aerosols are an important factors influencing global climate (Bonasoni et al., 2010; IPCC, 2013). They are the major culprits in visibility degradation (Srivastava et al., 2008) and effective carriers of toxic chemicals such as polycyclic aromatic hydrocarbons (PAHs) (Kaur et al., 2013; Sarkar and Khillare, 2013). PAHs, a large group of organic compounds with two or more aromatic rings, are generated through incomplete combustion or pyrolysis of materials such as coal, oil, gas, garbage, and biomass (Liu et al., 2007; Nielsen et al., 1996; Rajput et al., 2014). PAHs have received considerable attention for their potential risk to both human health and ecosystem stability (Bhargava et al., 2004). Since some PAHs (e.g. 4-6 ring PAHs) are persistent in the atmosphere, they can be transported over long distances to such remote areas as the Arctic or the Himalayas, thousands of kilometers away from their sources (Crimmins et al., 2004; Ding et al., 2007; Rajput et al., 2013). Dry and wet depositions of PAHs are major transport processes for atmospheric PAHs to aquatic and terrestrial surfaces. The study of PAHs in remote sites is therefore needed for the understanding of the atmospheric mechanisms involved in the long-range transport of these pollutants.

The Himalayas are a remote and mountainous region where emissions from anthropogenic activities are not significant when compared to the upwind adjacent regions in South Asia such as the highly populated, rapidly developing, and polluted Indo–Gangetic plains (IGP). The IGP is a major regional/global emitter of organic pollutants into the atmosphere (Singh et al., 2013). For example, Delhi and Kanpur, two large cities in the IGP, exhibit very high PAH concentrations, with annual mean values of the sum of PAHs of 850.2 ng/m$^3$ and 660.8 ng/m$^3$, respectively (Masih et al., 2010; NEERI, 2006). It has been estimated that southern Asia produced about 90 Gg/y of PAHs in 2004 (Zhang and



Tao, 2009), and the transport of PAHs from the IGP is considered to be the most important pathway controlling the levels of PAHs over the Himalayas (Gong et al., 2014; Rajput et al., 2013). Increasing pollutant emissions associated with the fast–growing economies of South Asian countries have led to the progressive increase of aerosol concentrations above the natural baseline, with a clearly measurable

positive trend in the last 30 years (Gautam et al., 2009). The Asian monsoon circulation and westerly winds are responsible for the distribution and transport of air pollutants to the Himalayan region (Bonasoni et al., 2010). Annually, this region has a long (Nov–May) dry season with accumulated air pollutants, notably on the southern side of the Himalayan Mountains. The so-called "Asian Brown Cloud (ABC)", a 3 km thick brownish layer of pollutants, has been detected extending from the Indian

Ocean to the Himalayas (Ramanathan et al., 2007; Tripathi et al., 2005). The aerosols that gather in the foothills can be lifted to high altitudes (Decesari et al., 2010; Qiu, 2013) or even travel over the high Himalayas and enter into the Tibetan Plateau (TP) (Bonasoni, 2008; Lüthi et al., 2015; Xia et al., 2011), thus affecting the atmospheric quality of the Himalayas.

It is therefore important to understand the emission, transport, transformation, removal and impacts

of various air pollutants, including PAHs in the Himalayas. Although some studies on elemental and carbonaceous particle concentrations have been done (Bonasoni et al., 2010; Rajput et al., 2013), current knowledge of PAH compositions for this region remains deficient (Kishida et al., 2009; Rajput et al., 2013), especially for its spatial and temporal variations. Here, we present results of a year-long measurement of TSP and particulate-phase PAHs from six sites (Lumbini, Pokhara, Jomsom, Dhunche

in Nepal, and Zhongba and Nyalam on the TP) across two Himalayan transects to investigate characteristics of PAHs in this critical region (Fig. 1). This paper aims to not only provide a better understanding of the temporal and spatial characteristics of ambient TSP and PAH concentrations over the Himalayas but will also shed light on the mechanism of long-range transport of other particle-bound pollutants.



## 2 Sampling and analysis

### 2.1 Sampling site description

The TSP samples were collected along two south–north transects, extending from Lumbini to Zhongba and Dhunche to Nyalam across both sides of the central Himalayas (Fig. 1). Lumbini, which lies on the border of India and Nepal, is in a rural area of the IGP. The major pollution sources are from agricultural activities and house heating and cooking, from the burning of large amounts of crop straw and wood each year, especially in winter and the pre-monsoon season, as well as from eleven cement factories and more than fifty other industries along the nearby Lumbini–Bhairahawa industrial corridor. Pokhara, located about 100 km north of Lumbini, is Nepal's third largest city, with a metropolitan population exceeding 200,000. In recent years it has experienced rapid urbanization with increased numbers of vehicles. Jomsom, a small town in the Mustang district of Nepal, is located in the Kali Gandaki Valley–a river valley that cuts across the Himalayas at the transition point between South Asia and Central Asia. Major human activities are tourism and limited agriculture. The measurement station in Jomsom itself is around 100 m higher up, and away from the only road that runs along the valley. Since the location is remote with minimal local emissions, we consider the site to be, to the extent possible, a regional background in the Himalayas. Dhunche is situated in the Langtang National Park in the Rusuwa district, about 50 km north of Kathmandu and 14 km south of the Chinese border. It is a semi-urban small town and the headquarters of the Rusuwa district. The major sources for pollution are biomass burning, vehicle emission, tourism, and agricultural activities surrounding the town. Zhongba and Nyalam are located on the northern side of the Himalayas, and are characterized by yak husbandry. The sampling site in Zhongba is around 20 km away from the town with a minimum influence of local emission in the atmosphere. Nyalam is a town with some local emissions including biomass combustion for cooking and heating and limited vehicular traffic.



## 2.2    Meteorology and backward air-mass trajectories

The weather differs from the southern to the northern sides. For sites in Nepal, the dominant surface wind direction is from the south during the monsoon season. In other seasons, northwesterly winds prevail due to the effect of the westerly winds (Bonasoni et al., 2010). For sites on the TP, southwesterly winds near the surface dominate during the monsoon season, while northeasterly winds prevail in other seasons (Ma et al., 2011). All sites experience four distinct seasons: winter (December to February), pre-monsoon (March to May), monsoon (June to September), and post-monsoon (October to November) (Bonasoni et al., 2010).

To reveal the transport pathway of air masses that arrive at sampling site on the north side of Himalayas, five-day air mass backward trajectories were calculated using the Hybrid Single-particle Lagrangian Integrated Trajectory (HYSPLIT) model and Global Data Assimilation System (GDAS) data provided by the Air Resources Laboratory of National Oceanic and Atmospheric Administration (http://www.arl.noaa.gov/ready/hysplit.html). The cluster trajectories were calculated at the height of 500 m starting at 00:00 Coordinated Universal Time on the sampling day.

## 2.3    Sample collection

A total of 190 aerosol samples were collected at the six sites during the period of April 2013 to March 2014. They were collected on pre-burned (550 ℃, 6 h) quartz fiber filters (90 mm in diameter, Whatman plc., Maidstone, UK.) using six samplers fitted with TSP cyclone (flow rate: 100 L/min, KC-120H: Qingdao Laoshan Applied Technology Institute, Qingdao City, China). Samples were collected every three to seven days for 24-h periods at Lumbini, Pokhara, Dhunche, and Jomsom and 48-h periods at Zhongba and Nyalam. The samplers were set up on rooftops at varying heights of 2-10 m above the ground (Table 1). Field blank filters were placed in the sampler for 24-h with no air drawn





through it. In some periods, such as during monsoon season, several samples could not be collected due to equipment breakdown associated with frequent precipitation or electricity supply failures.

The pre- and post-sampling weights of all quartz filters were measured with a microbalance, after equilibration at constant temperature and humidity (20 ℃, 39%) for at least 24-h. The samples were stored at -20 ℃ prior to extraction and chemical analysis. Before analysis, 1000 ng of hexamethyl benzene was added as internal standards. The volume of air passing through each filter was converted into standard conditions using atmospheric pressure and ambient temperature monitored at each site.

## 2.4 Extraction and analysis

Detailed procedures of ultrasonic extraction and analysis have been described in an earlier study (Wang et al., 2013). The PAHs were analyzed at the State Key Laboratory of Cryospheric Sciences, Cold and Arid Regions Environmental and Engineering Research Institute in Lanzhou, China using gas chromatography−mass spectrometry with a $30 \times 250$-μm ID HP–5MS capillary column following the method described in a previous publication (Chen et al., 2015). Further information about the extraction and analysis method is provided in Text SI-1. Samples were analyzed for the 15 different PAHs listed by the US EPA as priority pollutants: acenaphthylene (Acel), aecnaphthene (Ace), fluorine (Flu), phenanthrene (Phe), anthracene (Ant), fluoranthene (Fla), pyrene (Pyr), benzo(a)anthracene (BaA), chrysene (Chr), benzo(a)pyrene (BaP), benzo(b)fluoranthene (BbF), benzo(k)fluoranthene (BkF), dibenzo(a,h)anthracene (DahA), benzo(g,h,i)perylene (BghiP) and indeno(1,2,3-cd)pyrene (IndP). Naphthalene (Nap) was not analyzed because it was detected with high concentration in the laboratory and field blanks.



## 2.5 Quality control (QC)

All analytic procedures were carried out under strict quality assurance and control measures. Details regarding quality assurance procedures are given in Text SI-2. Laboratory and field blanks were extracted and analyzed in the same way as the samples. The detection limits were 0.84 (Acel), 0.59 (Ace), 0.70 (Phe), 0.39 (Ant), 0.06 (Flu, Fla, Bbf, DahA, and BghiP) and 0.03 pg/m$^3$ (Pyr, BaA, Chr, Bkf, BaP, and IndP). The recoveries in field samples were 74-93%, 80-97%, 83-105%, and 89-109% for acenaphthene-d10, phenanthrene-d10, chrysene-d12, and perylene-d12, respectively. In the field bland samples, Aecl, Ace, Phe, and Ant were detected to be 0.59, 0.45, 0.68, and 0.36 pg/m$^3$. The PAH concentrations were blank corrected but not corrected for the recoveries.

## 3 Results and discussion

### 3.1 Spatial distribution and seasonal trend of TSP and total PAH concentrations

The TSP and total PAH levels were remarkably different among the six sampling sites with a clear decreasing trend from the southern to the northern side of the Himalayas (Table 2). ANOVA－test showed that the concentrations of six sites were significantly different at the 0.05 level. Among the six sites, Lumbini was found to have the highest TSP and PAH concentrations (TSP: 209 ± 113 μg/m$^3$; PAHs: 91.6 ± 54.6 ng/m$^3$). These values were similar to what has been reported in many other South Asian cities, such as Delhi (Sarkar and Khillare, 2013) and Agra (Rajput and Lakhani, 2010). Going north from Lumbini, the annual average TSP and PAH concentrations of Pokhara decreased to 123 ± 98.1 μg/m$^3$ and 20.7 ± 14.7 ng/m$^3$, respectively. Pokhara has experienced increased urbanization and increased numbers of vehicles in recent decades due to increased tourists and local residents. It is also a region with significant rural biomass burning partially insulated from the IGP surface air by mountains



situated between Lumbini and Pokhara that might act as potential barriers to polluted air mass transport. Both Dhunche and Jomsom are located in rural areas of the central Himalayas. However, the TSP and PAH concentrations of Dhunche (TSP: $133 \pm 73.7$ μg/m$^3$; PAHs: $18.6 \pm 5.65$ ng/m$^3$) were similar to those of Pokhara due to relatively intensive local anthropogenic activities. TSP and PAH concentrations

of Jomsom (TSP: $96.2 \pm 40.8$ μg/m$^3$; PAHs: $11.1 \pm 2.97$ ng/m$^3$) were found to be comparable with those of Barapani (PAHs: $14.1$ ng/m$^3$), located in the foothill of the Himalayas, which is clearly impacted by transport of pollutants from the IGP in winter (Rajput et al., 2013). Considering that local emissions in Jomsom are very low, the measured pollutants probably accumulate due to the Kali Gandaki Valley's very fast up-valley winds that likely transported pollutants from central Nepal and the IGP. The lowest

TSP and PAH concentrations were observed in Zhongba and Nyalam, located on the northern side of the Himalayas, with average concentrations comparable with earlier reports from other sites on the TP (Wang et al., 2014) and from background regions of Central Europe (Dvorska et al., 2012).

The time series of TSP and PAH concentrations from Lumbini and pokhara in Nepal showed a clear seasonal variation, with high concentrations in the pre-monsoon season, gradually decreasing to

minimum concentrations around mid-monsoon season, and gradually increasing through post-monsoon to the maximum concentrations in winter season (Fig. 2). In other sites, the seasonal variation of TSP and total PAH concentrations were slightly higher during the non-monsoon season than those in the monsoon season. Intriguingly, the total PAH concentrations in Nyalam and Dhunche had peak values in the pre-monsoon season of 2013 which might be associated with pollution event. ANOVA－test shows

that the seasonal variation of each site was not significantly different at the 0.05 level. There are a variety of sources of TSP and PAHs in Nepal. For example, biomass such as agricultural wastes, animal dung, and wood, is burnt almost throughout the year mainly for cooking activities. Burning of large amounts of agro-residue also occurs in the IGP and Nepal, especially in the non-monsoon seasons (Ram



and Sarin, 2010), emitting a large amount of PAHs and other air pollutants (Sinha et al., 2014). Forest fires also increase significantly in the pre-monsoon season in this region (Vadrevu et al., 2012). Moreover, the unique weather conditions of this region (long dry season extending from November to May) combined with the geophysical features creates a conducive environment for the accumulation of

pollution in the air during the non-monsoon season. This leads to the formation of regional scale plumes of air pollutants, known as the ABC, resulting in high concentrations of pollutants on the southern side of the Himalayas (Ram et al., 2012; Rengarajan et al., 2007). Thus, the increase in TSP and PAH concentrations in the non-monsoon season is likely due to the combined effects of source strength and lower boundary layer height (Datta et al., 2010). In contrast, high rainfall during the monsoon season

washes out most of the PAHs and other pollutants from the atmosphere. Although the amplitude of variation between monsoon and non-monsoon seasons of TSP and PAH concentrations decreased significantly along these two south-north transects, both sites in Tibet exhibited similar seasonal variation to those in Nepal, suggesting that the region, at least the northern side of the Himalayas, is affected by long-range transported pollution.

## 3.2    Altitudinal dependence of TSP and PAH concentrations

In addition to the distance to the IGP, altitude is another important factor influencing the distribution of TSP and PAHs in the study region (Chen et al., 2008). Although several studies have investigated TSP and PAHs concentrations in the IGP (Rajput et al., 2013; Ram et al., 2010), these investigations mainly focused on single sites especially those in urban/rural regions, and they paid little attention to the PAHs

that are good markers for long-range transported pollutants.

The TSP and PAH concentrations exhibited altitudinal gradients in study region (Fig. 3). For example, the annual average TSP and PAH concentrations decreased by approximately 16 and 3 times of





concentrations from the lowest to the highest altitude sites. Representation of the PAH and TSP concentrations vs. altitude shows a logarithmic decreasing trend with statistically significant correlation coefficients, especially in the non-monsoon seasons (TSP: y= -57.3lnx + 552, $R^2$ = 0.952; PAHs: y= -26.8lnx + 229, $R^2$ = 0.948) (Fig. 3). Previous studies have reported similar altitudinal distributions of persistent organic pollutants in remote mountain areas, either in air, soil, or plants. For example, Gong et al. (2014) observed that the concentrations of such POPs as HCB and PCBs decreased approximately 3 times from 135 m to 5100 m. Guzzella et al. (2011) reported that PAH concentrations in the soil showed a significant negative correlation with altitude (2730-5293m) at Mt. Sagarmatha, Himalayas. The low–elevation sites displayed higher PAH concentrations, as they are strongly affected by the nearby contaminant sources. The high–elevation sites showed lower values because regional atmospheric circulation transported pollutants from lowland source regions predominate at these sites. Thus, less local anthropogenic emissions at higher elevations and dry and wet deposition from long-range transport should be the main reason for the decrease in concentrations of these compounds. However, it should be noticed that Zhongba and Nyalam are located on the north side of the Himalayas, thus we just gave a rough estimate of the regression analysis in this study.

### 3.3 Composition of PAHs

PAH profiles are most likely influenced by the emission source (Liu et al., 2007; Xu et al., 2012) and the amount of photo degradation due to their different atmospheric reactivity in the atmosphere (Butler and Crossley, 1981; Ding et al., 2007). The half-lives of PAHs vary significantly with higher half-lives of Phe, Fla, Chr, and Pyr than BghiP, IndP, Ant, and BaP (Behymer and Hites, 1985; Butler and Crossley, 1981; Kamens et al., 1988). Therefore, the composition profile of PAH would be influenced by the transport distance to some extent.



Fig. 4 shows the percentage contribution of PAHs with different rings to total concentrations at each site. For Lumbini, the major contributors to the total PAHs were 5- and 6-ring PAHs (Fig. 4), with contributions to be 38.6% and 39.1%, respectively (Table SI-1). High molecular compounds with low vapor pressure significantly dominated the PAH composition profile, indicating that the particulate

PAHs in Lumbini derive directly from the surrounding continents through short-range transport with less photo-degradation (Li et al., 2006; Liu et al., 2007). In addition, this profile pattern was consistent with the aerosol samples collected from post-harvest biomass burning emissions in the IGP (Rajput et al., 2011), indicating that biomass combustion is the the main source for particulate PAHs in Lumbini.

Moving north to Pokhara, the pattern was different with that of Lumbini. It had higher contributions

of 3-ring (17.3%) and 4-ring (29.3%) PAH when compared to Lumbini. It was found proportion of high molecular weight PAHs to total PAHs decreases significantly for samples in the foothills of the Himalayas, with the exception of Dhunche, suggesting that the important contribution of local emissions at Dhunche (Fig. 4). Other three background sites, including Jomsom, Zhongba, and Nyalam, showed relatively similar contributions of 4 groups to total PAHs (Fig. 4 and Table SI-1). Generally, the

majority of high molecular weight PAHs is present in the particulate phase due to their low vapor pressures. As a result, high molecular weight PAHs scarcely reenter the atmosphere after scavenging by the intensive dry/wet deposition, which occurs frequently during long-range atmospheric transport (Simonich and Hites, 1995; Wania and Mackay, 1996). Thus, the long distance to emission sources from these three remote sites should be the reason for their similar contribution profile.

Wang et al. (2014) have reported the composition of PAHs in soil and air samples across the inland of TP which showed that the atmospheric PAHs dominated with 3-ring PAHs (80%). This is different from our study (28.6% and 25.2% for Zhongba and Nyalam, respectively). In addition, burning of yak dung for domestic activities is common in populated areas across the TP (Li et al., 2012). However, the





PAH profiles in Zhongba and Nyalam were much different from that of yak dung combustion aerosol of the TP which had high concentrations of 3- and 4-ring PAHs (e.g. Phe, Fla, Pyr, and Chr ) (Li et al., 2012). In contrast, the PAHs profile in Zhongba and Nyalam were similar to that of Jomsom (Fig. 4), which might also reflect the long-range transport of pollutants from south to north side of the Himalayas.

## 3.4 Dry deposition fluxes estimation

Dry deposition fluxes of PAHs can be estimated from their ambient concentrations through Eq. (1) (Terzi and Samara, 2005):

$$F = C \times V_d \qquad (1)$$

where $V_d$ is the deposition velocity, C is the particle phase PAH concentration. Although no previous measurement of PAH dry deposition velocities in this area was available for calculation. We can estimate the dry deposition fluxes using the velocities reported by previous studies. The greatest uncertainty in this approach stems from the selection of a proper deposition velocity because the velocities for total PAHs varied largely from 0.01 to 6.7 cm/s among different areas. For example, Bodnar and Hlavay (2005) calculated $V_d$ values between 0.01 and 0.28 cm/s on the Lake Balaton. Bozlaker et al. (2008) found that the overall average $V_d$ was 2.9±3.5 cm/s in an industrial region in Turkey. Odabasi et al. (1999) have reported $V_d$ values ranged from 4.3 to 9.8 cm/s with an average of 6.7 cm/s in urban Chicago. However, the dry deposition velocities were similar to 1.4 cm/s for particle phase PAHs. For example, Demircioglu et al. (2011) have reported that the overall average deposition velocities for particle phase PAHs were 1.5±2.4 and 1.0±2.3 cm/s for suburban and urban sites in Turkey. Sheu et al. (1996) determined the mean dry deposition velocity of PAHs as 1.03 and 2.32 cm/s in the urban site and industry region in Taiwan. McVeety and Hites (1988) reported that the dry deposition velocities for particle-phase PAHs were 0.99 cm/s in Lake Superior. Thus, the $V_d$ was set as





1.4 cm/s in this study and the dry deposition fluxes of particle phase PAHs could be calculated using the above equation. Table 2 listed the dry deposition fluxes contributed by particle phase PAHs in the study region. In consistency to the ambient PAH levels, the particle phase PAH deposition fluxes were different among the six sampling sites with a clear decreasing trend from the southern to the northern

side of the Himalayas. Among the six sites, Lumbini was found to have the highest dry deposition fluxes (110.8 µg/m$^2$ d). Going north from Lumbini, the annual dry deposition fluxes of Pokhara decreased to 25.9 µg/m$^2$ d. Dhunche had higher dry deposition fluxes (22.5 µg/m$^2$ d) than that of Jomsom (13.4 µg/m$^2$ d) because of relatively intensive local anthropogenic activities. The lowest dry deposition fluxes were observed in Zhongba (10.6 µg/m$^2$ d) and Nyalam (6.74 µg/m$^2$ d). As previous

studies reported, the $V_d$ could be affected by meteorological parameters (wind speed, atmospheric stability, relative humidity) and physical characteristics of pollutant (particle size, shape). Consequently, $V_d$ can vary by orders of magnitude while the assumption of a constant value may introduce large uncertainties in the calculation of dry deposition fluxes (Seinfeld and Pandis, 1998). However, the rough estimated dry deposition fluxes reported in this study were useful in assessing the atmospheric

environment and its impacts on the Himalayan ecosystem.

## 3.5    Source identification

Parent PAH ratios are frequently used to identify the origin of PAHs (Rajput et al., 2014; Yunker et al., 2002). Generally, various ratios are used simultaneously to cross-check the results and to reduce uncertainties. In this study, the concentration ratios of IndP/(IndP+BghiP) and Fla/(Fla+Pyr) in the

particulate-phase were selected as indicators to investigate emission sources (Table 3). In general, a ratio of IndP/(IndP+BghiP) lower than 0.2 indicates non-burned petroleum as the primary input, a value between 0.2 and 0.5 indicates petroleum combustion, and values higher than 0.5 suggest that biomass and coal burning are the primary inputs; for Fla/(Fla+Pyr), a ratio lower than 0.4 probably implies



unburned petroleum, 0.4-0.5 implies petroleum combustion, and higher than 0.5 implies grass, wood or coal combustion (Yunker et al., 2002). In this study, the average IndP/(IndP+BghiP) ratio of Lumbini was found to be 0.55±0.037 during the sampling period, which was consistent with the mixed combustion of biomass and coal. In Lumbini, burning biomass such as wood and agricultural wastes for cooking occurs almost throughout the year. And large amounts of agro-residue (post-harvest burning of rice-residue in October-November and wheat-residue burning in April-May) also burned in the IGP and Nepal (Ram and Sarin, 2010). In addition, there are numerous of brick kilns near Lumbini, which is carried out from January to April. Thus, the combined industry (coal consume) and biomass (wood and residue) emissions are the primary sources. Correspondingly, the mean Fla/(Fla+Pyr) ratio was 0.49±0.026, similar to the corresponding value of wheat burning, implying that biomass, especially agro-residue burning, plays an important role in atmospheric PAHs at Lumbini.

The IndP/(IndP+BghiP) (0.68±0.096) and Fla/(Fla+Pyr) (0.42±0.113) ratios from Pokhara differed from those of Lumbini (Table 3) and were consistent with values from biomass, coal burning and petroleum combustion, respectively. There may be a local contribution from Pokhara city because the increased numbers of vehicles in recent decades due to increased tourists and local residents (with population of 200000). There also has significant rural biomass burning around the city. Thus, the PAHs in Pokhara were caused by the combined impact of local contributions from Pokhara city (both biomass and fossil burning) and polluted air masses from upwind over the IGP, which often arrive during the winter and pre-monsoon seasons.

The ratios of IndP/(IndP+BghiP) (0.52±0.023) and Fla/(Fla+Pyr) (0.48±0.029) from Dhunche were found to be similar to Lumbini's, indicating that Dhunche was also seriously affected by biomass and coal combustion as well as petroleum fuel combustion. However, the ratios of IndP/(IndP+BghiP) in Jomsom, Zhongba, and Nyalam were similar (0.44±0.021, 0.44±0.016, and 0.45±0.038, respectively)



with corresponding values of wheat burning. The same pattern was also found in the ratio of Fla/(Fla+Pyr) (0.47±0.013, 0.48±0.029, and 0.48±0.012, respectively), indicating that biomass burning might be a great contributor in these background regions. However, these two ratios at three remote sites were also in the range of the petroleum combustion, indicating that such liquid fuel combustion might be another source for PAHs. Since these three locations are remote with minimal local emissions, we consider these sites to be, to the extent possible, influenced by the pollutants from long-range transport.

## 3.6    Transport of PAHs across the Himalayas

The PAH concentrations at high mountain regions are generally influenced by long-range transport from nearby continental sources. China and India were identified as important sources of PAHs, contributing 30% (114 Gg/y) and 23.6% (90 Gg/y) of the total global emissions, respectively (Zhang and Tao, 2009). In addition, Pakistan was also one big emitter for PAHs (12 Gg/y). The study region was surrounded by these three countries. However, it is difficult to determine the potential source region and compare the relative strength of different sources to the sensitive mountain regions. Therefore, the HYSPLIT back trajectories were used to identify possible source regions and to explore the influence of long-range transport of aerosols to the northern side of the Himalayas (take Zhongba for example). These trajectories were consistent with other descriptions of atmospheric circulation patterns corresponding to the South Asian monsoon regime (Fig. 5) (Cong et al., 2009). In the monsoon season, most of the air masses come from the IGP (72%), and bring moisture and polluted aerosols. In the non-monsoon seasons, the transport pathways of air masses arriving at Zhongba were similar to the strong westerlies that pass through western Nepal, northwest India, and Pakistan. And the backward trajectories indicated that the IGP is the major source region. However, the results based on HYSPLIT back trajectories have large uncertainty due to the coarse resolution of their meteorological fields




(Koracin et al., 2011). In addition, the HYSPLIT trajectories do not take into account thermally driven flows through Himalayan valleys and up sides. Therefore,the topographic effect on air pollutant transport should also be considered. In mountainous areas, a diurnal valley wind system often occurs that blows up valley during the day and reverses to downward during the night. In this situation, a significant temperature difference exists between mountaintop and lowland. A previous study has reported that the wind regime at Nepal Climate Observatory–Pyramid, located on the southern side of the Himalayas, was characterized by an evident daily cycle of mountain/valley breeze (Bonasoni et al., 2010). During the daytime, the up-valley winds (southerly) with maximum wind speed in the afternoon can deliver the air pollutants from the foothills to higher altitudes. However, a different mountain–valley breeze circulation was observed on the northern side of the Himalayas, with a down-valley wind dominating in the daytime, especially in the afternoon. Therefore, the local mountain/valley breeze circulation acts as the connection for the air pollutants crossing the Himalayas (Cong et al., 2015). In addition, the big valleys between Jomsom and Zhongba (Mustang valley) and Dhunche and Nyalam (Langtang valley) provide potential channels for the transport of PAHs crossing the Himalayas (Lüthi et al., 2015; Marinoni et al., 2010; Xia et al., 2011).

## 4 Conclusions

This study investigated the PAH concentrations of atmospheric aerosols along two south-north Himalayan transects. The TSP and PAH concentrations decreased significantly from the southern to the northern side of the Himalayas, with the highest concentrations found in Lumbini (TSP: $210 \pm 113$ $\mu g/m^3$; PAHs: $91.6 \pm 54.6$ $ng/m^3$) and the lowest concentrations in Nyalam (TSP: $59.1 \pm 62.0$ $\mu g/m^3$; PAHs: $5.57 \pm 3.36$ $ng/m^3$). Similar to the ambient PAH levels, the highest and lowest dry deposition fluxes were also found in these two sites (Lumbini: $110.8$ $\mu g/m^2$ d; Nyalam: $6.75$ $\mu g/m^2$ d). In addition



to latitude, the PAH and TSP concentrations also exhibited a logarithmic decreasing pattern with increasing elevation, especially in non-monsoon seasons (TSP: y= -57.3lnx + 552, $R^2$ = 0.952; PAHs: y= -26.8lnx + 229, $R^2$ = 0.948). Moreover, the PAH and TSP concentrations showed a clear seasonal cycle, with higher concentrations during the non-monsoon season than those in the monsoon season,

reflecting the different transport and dispersion of emissions. In addition, the high percentage of 4-6 ring PAHs in Lumbini, Pokhara, and Dhunche indicated that local emissions play an important role in the accumulation of pollutants, while the increase of 3-ring PAHs at the other three remote sites, reflecting the higher deposition efficiency of high molecular weight PAHs during long-range transport. The evaluation of diagnostic molecular ratios indicated that atmospheric PAHs in Nepal originate

mainly from the use of coal, biomass fuels and vehicular emissions. Based on the backward air-mass trajectories, we found that the IGP is the potential source regions of the northern side of the Himalayas. In addition to large-scale atmospheric circulation, the unique mountain/valley breeze system in the Himalayas could also have an important effect on air pollutant transport between the two sides of the Himalayas.

## Appendix A: Mathematical background

Text SI-1 Sample extraction and analysis: A quarter of each filter was cut into pieces, placed into a glass tube, and immersed in 20 mL of dichloromethane (DCM) and *n*-hexane (1:1). The extraction was performed by sonication twice for 30 min at 27 ℃. Every single sample was spiked with deuterated

PAHs (naphthalene-d8, acenaphthene-d10, phenanthrene-d10, chrysene-d12, and perylene-d12) as recovery surrogates. The extracts were evaporated to about 0.5 mL with a rotary evaporator, and transferred to a multilayer column filled with 2 g of activated silica gel, 4 g of neutral alumina, and 1 cm of anhydrous $Na_2SO_4$ (pre-soaked in *n*-hexane). Then the column was eluted by a mixture of 10 mL



of *n*-hexane and 20 mL of DCM/*n*-hexane (1:1). The eluent solvent was blown down to a final volume of 1 mL under a gentle stream of nitrogen. Finally, the solution was transferred to a 1.5-mL vial and stored at -20 ℃ for rejection.

High-purity helium was used as a carrier gas at a constant flow rate of 1.0 mL/min. The mass spectrometer was operated in 70-Ev electron impact mode. The oven temperature was held stable at 100 ℃ for 2 min, then was increased to the final temperature of 260 ℃ at different rates; to 170 ℃ at 25 ℃/min, to 225 ℃ at 8 ℃/min, to 235 ℃ at 0.7 ℃/min, to 260 ℃ at 25 ℃/min, and held at a final temperature of 260 ℃ for 2 min. The temperature of the injector was 250 ℃ and that of the transfer line was 280 ℃.

Text SI-2 Quality control: Laboratory blanks for air samples were included at a rate of one for every five samples and were treated in exactly the same manner as the samples. The field blanks for the air samples were extracted and analyzed in the same way as the samples. Method detection limits (MDLs) were derived as 3 times the standard deviation of the mean blank concentrations. Except for Nap, most of the PAHs were not detected in the laboratory and field blanks, indicating contamination of most PAHs was negligible during transport, storage, and analysis. Therefore, we decided to not analyze the Nap component from the field samples. The concentration of the lowest calibration standard was taken as the detection limit (0.13 ng/sample for air samples).

Table SI-1 Percentage contribution (%) of each species to total PAHs in the atmosphere over the Himalayas

| PAH | Lumbini | Pokhara | Jomsom | Zhongba | Dhunche | Nyalam |
|---|---|---|---|---|---|---|
| Ace | 0.47 | 2.07 | 3.70 | 3.81 | 2.65 | 3.24 |
| Acel | 0.57 | 2.02 | 4.41 | 4.54 | 2.60 | 3.81 |





| | | | | | |
|------|------|------|------|------|------|
| Flu | 1.46 | 4.33 | 7.92 | 7.90 | 5.04 | 6.26 |
| Phe | 2.56 | 5.54 | 7.35 | 5.90 | 5.54 | 5.58 |
| Ant | 0.84 | 3.35 | 6.42 | 6.44 | 4.35 | 6.26 |
| 3-ring | 5.90 | 17.3 | 29.8 | 28.6 | 20.2 | 25.2 |
| Fla | 3.80 | 7.71 | 7.54 | 10.1 | 7.89 | 9.06 |
| Pyr | 3.88 | 7.77 | 8.30 | 9.32 | 8.28 | 9.77 |
| BaA | 3.72 | 5.50 | 5.47 | 6.42 | 5.99 | 5.98 |
| 4-ring | 16.4 | 29.3 | 30.3 | 36.2 | 30.9 | 35.4 |
| Chr | 5.03 | 8.36 | 8.95 | 10.3 | 8.72 | 10.5 |
| Bbf | 11.5 | 10.9 | 8.06 | 9.80 | 9.58 | 11.4 |
| Bkf | 12.0 | 11.8 | 6.11 | 6.99 | 10.5 | 3.50 |
| Bap | 12.9 | 8.72 | 7.03 | 7.28 | 7.80 | 7.29 |
| DahA | 2.26 | 3.41 | 5.02 | 2.26 | 4.22 | 3.75 |
| 5-ring | 38.6 | 34.9 | 26.2 | 26.3 | 32.1 | 25.9 |
| IndP | 22.5 | 9.72 | 6.21 | 3.35 | 8.77 | 5.91 |
| BghiP | 16.6 | 8.79 | 7.53 | 5.54 | 8.10 | 7.66 |
| 6-ring | 39.1 | 18.5 | 13.7 | 8.88 | 16.9 | 13.6 |

## Acknowledgements

This study was supported by the "Strategic Priority Research Program (B)" of the Chinese Academy of Sciences (XDB03030504), and the National Natural Science Foundation of China (41271015, 41225002, and 41171398). The authors gratefully acknowledge the efforts made by ICIMOD and





SusKat project in maintaining these sampling sites. We also thank university of Virginia and Nepal Wireless for running the Jomsom and Pokhara sites. In addition, the staff at the Nepal (Bohjraj Bhatta, Anil Patel, Gupta Giri, Buddhi Lamichhane, and Buddha Lama) and Tibet (Chaojun Xie and Nima Jiala) sampling sites helped enormously with this study.

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



Table 1. Location and description of six sampling sites.

| Sampling site | Location, Altitude | Setting | Details (height above the ground) |
|---|---|---|---|
| Lumbini | 27 °29′ N, 83 °17′ E, 100 m | Agricultural-Rural | Roof of a water tower (10m) |
| Pokhara | 28 °11′ N, 83 °59′ E, 813 m | Urban city | Roof of a residential house (6m) |
| Jomsom | 28 °46′ N, 83 °43′ E, 3048 m | Remote, background station | Roof of a laboratory (2m) on a nearby mountain |
| Zhongba | 29 °42′ N, 83 °59′ E, 4704 m | Remote, background country | Roof of a residential house (4m) |
| Dhunche | 28 °7′ N, 85 °18′ E, 2051 m | Semi-urban town | Roof of the regional environmental protection office (4m) |
| Nielamu | 28 °10′ N, 85 °59′ E, 4166 m | Semi-urban town | Roof of a residential house (6m) |





Table 2 Statistical summary of PAHs (ng/m$^3$), dry deposition fluxes ($\mu$g/m$^2$d) and TSP ($\mu$g/m$^3$) in six sites for a sampling period (April 2013-March 2014). The number in the parenthesis represents the standard deviation of individual PAH.

|        | Ring | Lumbini    | Pokhara    | Jomsom     | Zhongba    | Dhunche    | Nyalam     |
|--------|------|------------|------------|------------|------------|------------|------------|
| Ace    | 3    | 0.43(0.07) | 0.43(0.03) | 0.41(0.09) | 0.33(0.11) | 0.49(0.09) | 0.18(0.04) |
| Acel   | 3    | 0.52(0.22) | 0.42(0.05) | 0.49(0.11) | 0.40(0.08) | 0.48(0.12) | 0.21(0.04) |
| Flu    | 3    | 1.34(0.87) | 0.89(0.15) | 0.88(0.18) | 0.69(0.13) | 0.94(0.26) | 0.35(0.07) |
| Phe    | 3    | 2.35(1.64) | 1.14(0.45) | 0.82(0.32) | 0.52(0.10) | 1.03(0.47) | 0.31(0.15) |
| Ant    | 3    | 0.77(0.17) | 0.69(0.05) | 0.71(0.12) | 0.57(0.19) | 0.81(0.18) | 0.35(0.06) |
| Fla    | 4    | 3.48(1.76) | 1.59(1.34) | 0.84(0.22) | 0.89(0.74) | 1.46(0.72) | 0.50(0.42) |
| Pyr    | 4    | 3.56(1.82) | 1.61(1.26) | 0.92(0.22) | 0.82(0.55) | 1.54(0.64) | 0.54(0.43) |
| BaA    | 4    | 3.40(2.34) | 1.14(1.23) | 0.61(0.15) | 0.56(0.46) | 1.11(0.43) | 0.33(0.30) |
| Chr    | 4    | 4.61(2.86) | 1.73(1.94) | 1.00(0.19) | 0.91(0.57) | 1.62(0.59) | 0.59(0.39) |
| Bbf    | 5    | 10.5(7.90) | 2.26(2.85) | 0.90(0.29) | 0.86(0.53) | 1.78(0.57) | 0.63(0.64) |
| Bkf    | 5    | 11.0(7.99) | 2.44(3.05) | 0.68(0.56) | 0.61(0.48) | 1.94(0.57) | 0.19(0.38) |
| Bap    | 5    | 11.8(9.32) | 1.80(1.94) | 0.78(0.18) | 0.64(0.31) | 1.45(0.47) | 0.41(0.25) |
| IndP   | 6    | 20.6(15.1) | 2.01(1.36) | 0.69(0.24) | 0.29(0.27) | 1.63(0.62) | 0.33(0.36) |
| DahA   | 5    | 2.07(1.00) | 0.70(0.13) | 0.56(0.23) | 0.20(0.28) | 0.78(0.16) | 0.21(0.17) |
| BghiP  | 6    | 15.2(9.33) | 1.82(1.05) | 0.84(0.23) | 0.49(0.30) | 1.50(0.50) | 0.43(0.28) |
| T-PAHs |      | 91.6(54.6) | 20.7(14.7) | 11.1(2.97) | 8.78(4.46) | 18.6(5.65) | 5.57(3.36) |





| Flux | 110.8 | 25.0 | 13.4 | 10.6 | 22.5 | 6.74 |
|---|---|---|---|---|---|---|
| TSP | 209(113) | 123(98.1) | 96.2(40.8) | 67.8(32.1) | 132 (73.7) | 59.1(62.0) |
| Sample number | 37 | 30 | 30 | 24 | 37 | 32 |



Table 3 Diagnostic ratios of PAHs in aerosols of six sampling sites and their source profiles.

| | Lumbini | Pokhara | Jomsom | Zhongba | Dhunche | Nyalam | Source profiles |
|---|---|---|---|---|---|---|---|
| IndP/(IndP+BghiP) | 0.55±0.037 | 0.68±0.096 | 0.44±0.021 | 0.44±0.016 | 0.52±0.023 | 0.45±0.038 | <0.2 Petrogenic |
| Pre-monsoon | 0.57 | 0.75 | 0.45 | 0.42 | 0.52 | 0.45 | 0.2-0.5 Petroleum combustion |
| Monsoon | 0.53 | 0.60 | 0.43 | 0.42 | 0.54 | 0.43 | |
| Post-monsoon | 0.56 | 0.73 | 0.44 | 0.44 | 0.52 | 0.47 | 0.43 Wheat |
| Winter | 0.59 | 0.77 | 0.47 | 0.46 | 0.49 | 0.46 | 0.49 Rice |
| | | | | | | | >0.5 Grass, wood and coal combustion |
| Fla/(Fla+Pyr) | 0.49±0.026 | 0.42±0.113 | 0.47±0.013 | 0.50±0.035 | 0.48±0.029 | 0.48±0.012 | <0.4 Petrogenic |
| Pre-monsoon | 0.49 | 0.38 | 0.47 | 0.48 | 0.48 | 0.48 | 0.4-0.5 Petroleum combustion |
| Monsoon | 0.48 | 0.42 | 0.46 | 0.47 | 0.48 | 0.47 | |
| Post-monsoon | 0.51 | 0.46 | 0.47 | 0.49 | 0.48 | 0.48 | 0.46 Rice |
| Winter | 0.50 | 0.44 | 0.48 | 0.51 | 0.48 | 0.49 | 0.49 Wheat |
| | | | | | | | >0.5 Grass, wood and coal combustion |

Note: These source profile data were cited from Yunker et al. (2002); Rajput et al. (2011); Kavouras et al. (2011).





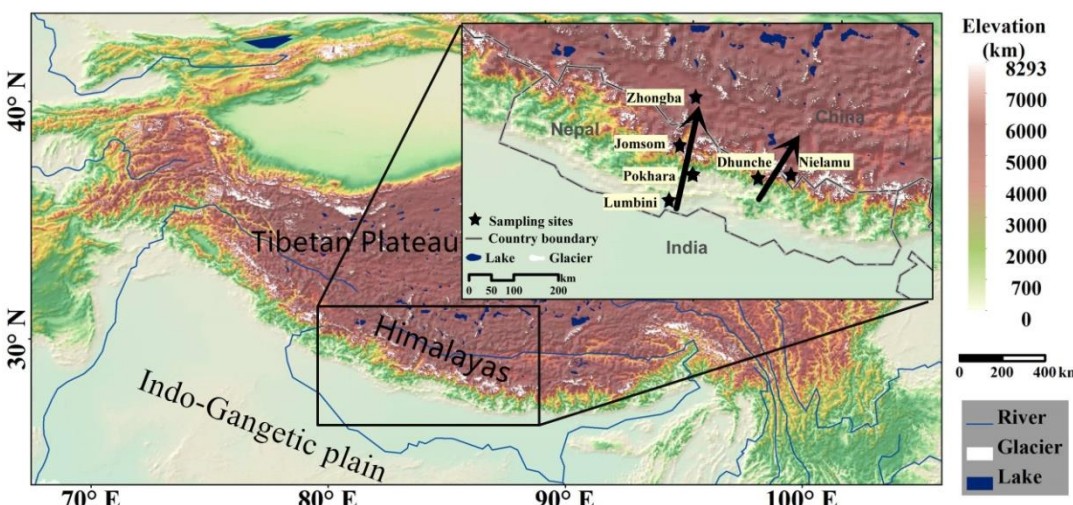

Figure 1. Location of the study area and sampling sites.



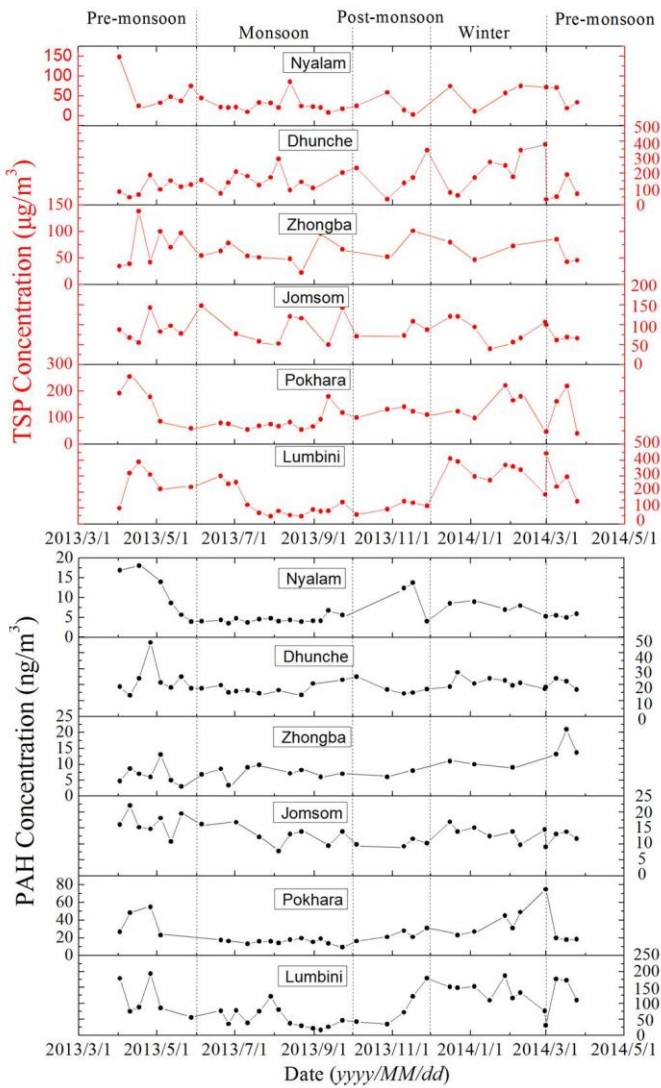

Figure 2. Seasonal variations of TSP ($\mu$g/m$^3$) and PAH (ng/m$^3$) concentrations of six sampling sites.





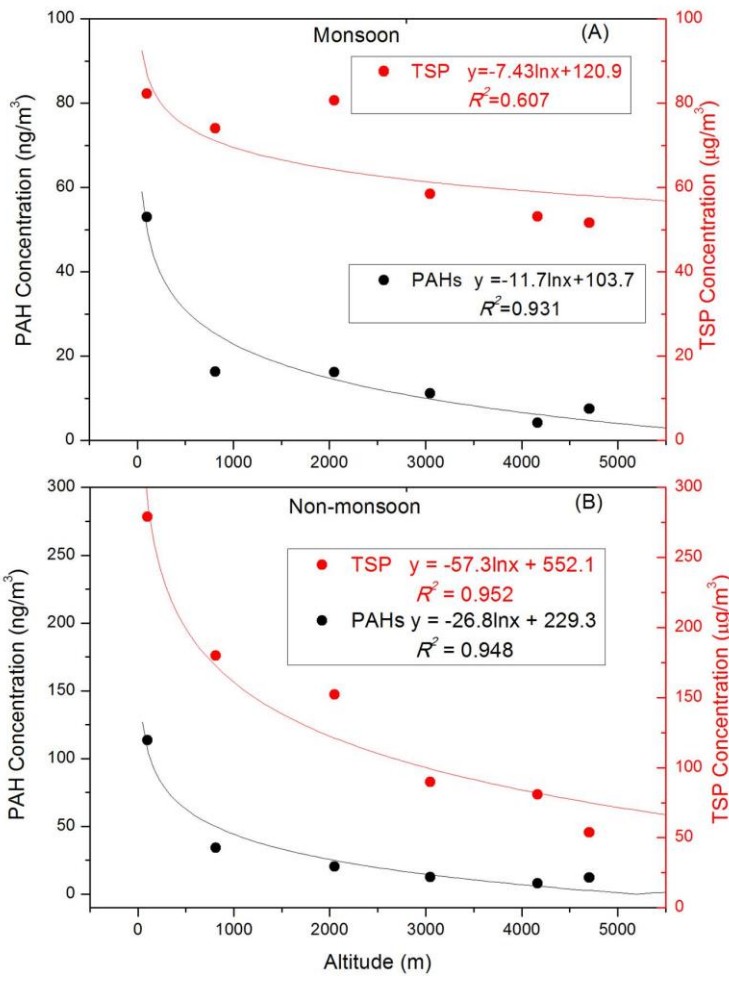

Figure 3. Concentrations of TSP ($\mu g/m^3$) and PAHs ($ng/m^3$) along the altitudinal gradient in the monsoon (A) and non-monsoon (B) seasons.



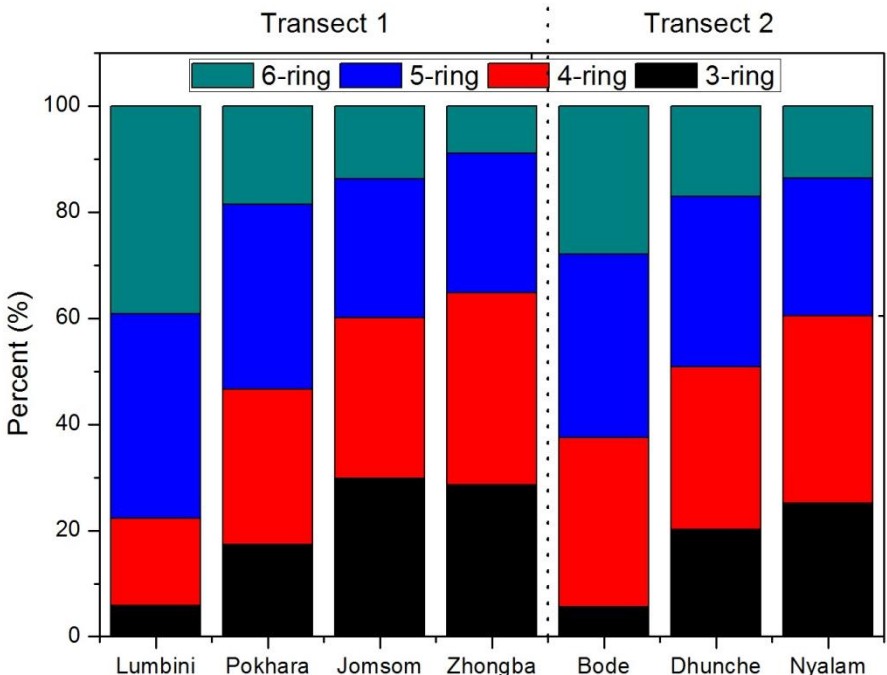

Figure 4. Percentage composition of 3, 4, 5, and 6 ring PAHs in aerosols of six sampling sites and Bode
(Kathmandu). Data from Bode site were reported by Chen et al. (2015)





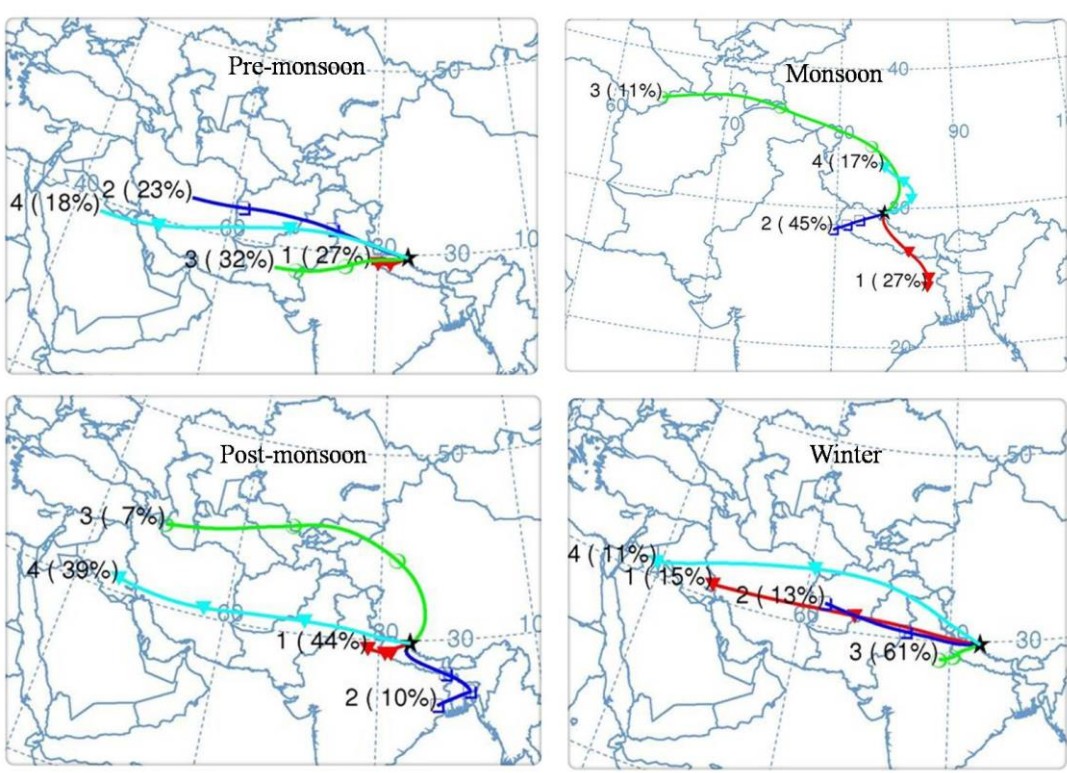

Figure 5. The cluster means 3-day backward air trajectories for Zhongba during different seasons.