# Peer review of "Supporting Information of"

_Atmospheric Chemistry and Physics, 2016_

## Referee Comment (RC1) · Anonymous Referee #2 · 12 Apr 2016

General Comments

This is an interesting manuscript dealing with the measurement of particulate PAH concentrations in the Himalayan region across two transects going from Nepal into China Tibetan plateau. The manuscript has new collected data that permits further insights into the background contamination of Himalayan region and the long distance transport of particulate PAH pollution from the Indian subcontinent. In my opinion (from a non-born English speaker) the quality of the English in the paper needs to be improved. In the text there are some sections that also need improvement. Section 3.1 about the spatial distribution and seasonal trend of TSP and PAH concentrations is presented in a confusing way and some of the declarations about spatial and seasonal

trends are not evident when compared with concentration data in Tables and Figures. For example in lines 15-21, page 9, there is the statement of a seasonal variation of concentrations which seems to be contradicted by ANOVA tests. On lines 7-10, page 10, it is concluded that there is an increase in concentrations during the non-monsoon season which, at least for TSP, is not evident from inspection of Figure 2. Section 3.4 concerning the estimation of dry deposition fluxes also needs improvement. Why there was a choice of a dry deposition velocity of 1.4 cm/s for calculating the PAH dry deposition fluxes, when there is a so large variability in measured fluxes? Is it an average of bibliographic data?

Specific Comments

Figure 1. The background site in China it is expressed as Nielamu while in the text the local is referred as Nyalam – correct! Abstract- IGP is only defined lately in the text. Introduce definition here. Page 6 line 15-20 – Define the size cutting characteristics of the TSP cyclone Page 8, line 7- change "bland" to "blank" Page 9, line 13- change "pomara" into "Pomara" Page 9, line 16- change "variation" to "variations" Page11, line 6- define POPs, HCBs and PCBs Page 12, line 6- change "continents" to "emission sources" Page13, line 14- this is an estimation/calculation and not a measurement. Figure 4- The site Bode with data taken from Chen et al., (2015) is presented in the figure but not discussed in the text. Please discuss.

---

## Referee Comment (RC2) · Anonymous Referee #1 · 18 May 2016

Review of acp-2016-71

General comments:

1) Overall, the manuscript is poor in English, making it difficult at several places to understand what is being conveyed. Section on estimation of dry-deposition fluxes of PAHs is irrelevant. It is not clear why it is important to estimate PAHs fluxes at the six sampling sites. The impact on aquatic systems, if any, or human health has not been addressed at all. Simply reporting the fluxes (rough estimates – as claimed by authors themselves on page 14, lines 14-15) do not make any sense. With large uncertainty in the deposition velocity of any atmospheric constituent, it is conceptually wrong to state the deposition fluxes of PAHs up to 1st or 2nd decimal units (25.9 ug/m2/d at Pokhara,

at Nyalam as 6.74 ug/m2/d and so on – Section 3.4, Pages 13-14). The entire concept of using deposition velocity and deposition fluxes is not valid at all for sampling sites located in high altitude regions. The concept may still be applicable for deposition over high altitude lakes.

2) Abstract is still very poorly written: Page 2, lines 1-2: 1) Why it is important to understand transport of PAHs across Himalayas? 2) Why only from the Indo-Gangetic Plain?

3) Page 2, lines 20-24: Isomer ratios are expected to help in identifying the specific source signature, whereas, authors have stated that – quote "isomer ratios suggested that atmospheric PAHs from the Nepal sites were mainly associated with emission of biomass, coal burning and petroleum burning". This is a very qualitative statement.

4) Conclusion: Page 18, lines 10-15: If inferences are drawn only from AMBTs, then it is not of much relevance to measure chemical constituents (example PAHs). Their long-range transport cannot be considered "conservative".

5) Page 18, line 8: What authors mean by "higher deposition efficiency"? How this is built in the concept of using deposition velocity for PAHs? How deposition efficiency is assessed from the data presented in the manuscript.

6) Specific Comments: a) Abstract, Page 2, line 11: What is the concept of logarithmic decreasing pattern of PAHs with increasing elevation? Is this is an empirical relation only applicable to PAHs? b) Page 3, lines 14-15: Why study of PAHs in remote sites is needed for the understanding of the atmospheric mechanisms involved in the long-range transport of these pollutants? Which "atmospheric mechanisms" authors are referring to during long-range transport? c) Page 4, line 8: ABC is not "Asian Brown Cloud". It refers to "Atmospheric Brown Cloud". d) Page 5, lines 5-19: Why these sources are not important for the contribution of PAHs measured at the sis sampling sites? e) Page 9, lines 4-8: Based on TSP and PAHs concentrations, it is conceptually incorrect to conclude impact and transport of pollutants from the IGP in the winter.

What about contribution from intermediate/downwind sources. f) Page 9, lines 3-5: Std. deviation on PAHs cannot be stated as 5.65 ng/m3, 2.97 ng/m3 and so on. Are these significant up to 2nd decimal units. g) Page 10, lines 10-14: It is not clear what authors are trying to infer and convey. It is rather poor discussion on spatial and temporal variability of TSP and PAHs along south-north transects. h) Page 11, lines 7-8: Concentrations of PAHs in soils and variability with altitude is out of context and irrelevant. PAHs in soils cannot be assumed to be derived from atmospheric deposition. i) Page 11, lines 9-10: Which "nearby contaminant sources" authors are referring to? j) Page 11, line 12: "Thus, less local anthropogenic emissions —— ". What is "less"? k) Page 11, lines 12-15: The entire discussion is very qualitative and poorly written. l) Page 11, line 15: "–thus we just gave a rough estimate of the regression analysis in this study". What is the relevance of giving "rough estimate"? m) Page 12, line 8: "——- indicating that biomass combustion is the main source for particulate PAHs in Lumbini". How biomass combustion source can be inferred from particulate PAHs? What are the concentrations and ratio of OC and EC? n) Page 15, lines 16-19: There is no new understanding emerging from this qualitative discussion. o) Page 17, lines 1-2: What is "thermally driven flows through Himalayan alleys and up sides? p) Page 17, lines 3-4: "—— diurnal valley wind system often occurs that blows up valley–". What authors mean by "that blows up valley—- "?

7) There are several confusing & qualitative statements: Page 2, lines 22-24; Page 3, lines 14-15; Page 9, lines 7-10; Page 10, lines 10-15; Page 11, lines 6-7; 10-15; Page 13, lines 3-4; Page 14, lines 12-15; Page 15, lines 16-19

8) English errors: Abstract, Page 2, line 11: "exhibited"; wrong English; Page 3, line 1: "long-range transportation" Page 6, line 17: "pre-burned"; What is pre-burned? It should be "pre-combusted"! Page 8, line 2: "All analytic"; What is analytic? Page 10, lines 14 and 20: "long-range transported pollution"? Wrong English "transported"! Page 10, line 21: "—- concentrations "exhibited"—-. Page 11, line 9: "The low-elevation sites displayed—-"; displayed is not a correct word to use! Page 12, line 15: "PAHs are

present"; not is present. Page 12, line 16: "— PAHs scarcely reenter the atmosphere—-". Very poor English! Page 14, line 21: "—— indicates non-burned petroleum——". Non-burned is incorrect word.

---

## Editor Comment (EC1) · E. Stone (Editor) · 18 May 2016

In addition to the comments made by referees, please address the following comments and suggestions:

1. The review of prior studies of PAH in the Himalayas on page 4 line 15-16 is very terse and has overlooked several relevant publications that provide insight to PAH sources and seasonal variation in the Himalaya:

Chen, P. F., S. C. Kang, C. L. Li, M. Rupakheti, F. P. Yan, Q. L. Li, Z. M. Ji, Q. G. Zhang, W. Luo and M. Sillanpaa, 2015. Characteristics and sources of polycyclic aromatic hydrocarbons in atmospheric aerosols in the Kathmandu Valley, Nepal. Science of the

[Figure]

Total Environment 538, 86-92.

Kim, B. M., J. S. Park, S. W. Kim, H. Kim, H. Jeon, C. Cho, J. H. Kim, S. Hong, M. Ru-pakheti, A. K. Panday, R. J. Park, J. Hong and S. C. Yoon, 2015. Source apportionment of PM10 mass and particulate carbon in the Kathmandu Valley, Nepal. Atmospheric Environment 123, 190-199.

Stone, E. A., J. J. Schauer, B. B. Pradhan, P. M. Dangol, G. Habib, C. Venkataraman and V. Ramanathan, 2010. Characterization of emissions from South Asian biofuels and application to source apportionment of carbonaceous aerosol in the Himalayas. Journal of Geophysical Research-Atmospheres 115.

2. In light of the abovementioned source apportionment studies – there is evidence for multiple sources of PAH (e.g. coal, biomass, and fossil fuel use) in the Himalayas. The limitations of using PAH isomer ratios for source identification in the presence of multiple sources should be discussed.

3. Isomer ratios of PAH have also been utilized as a measure of atmospheric aging, particular photochemical degradation (see Bi et al. 2003). Use of the appropriate isomer ratios to track aging may be useful in establishing quantitative support to evaluate local versus long range transport of PAH.

Bi, X. H., G. Y. Sheng, P. Peng, Y. J. Chen, Z. Q. Zhang and J. M. Fu, 2003. Distribution of particulate- and vapor-phase n-alkanes and polycyclic aromatic hydrocarbons in urban atmosphere of Guangzhou, China. Atmospheric Environment 37 (2), 289-298.

4. I concur with referees concerns with the validity of dry deposition flux estimation. A "rough estimation" using an assumed, untested, and unexplained deposition velocity is not valid. The only conclusion drawn is that flux trends follow concentration, which is obvious from Equation 1 when assuming a fixed deposition velocity. The resulting data are not used "assess the atmospheric environment and its impacts on the Himalayan ecosystem" as noted on page 14 lines 14-15. Consequently, section 3.4 should be

removed from the manuscript.

5. In the abstract, a number of improvements are needed: a) clarify the importance of studying PAH in the Himalays ("understanding. . . remains limited" is too vague); b) the names of the sites (with their altitudes) should be listed following "Himalayas:" at line 5; c) define x and y in the equations at lines 12-13.

6. In the introduction (page 3 line 15) clarify what "atmospheric mechanisms" specifically need to be understood and why.

7. The motivation to study PAH should be justified and clarified in the introduction. PAH generally have low acute toxicity to humans, and their most significant endpoint is cancer.

8. The "sum of PAH" noted on page 3 line 22 is not operationally defined by the method of analysis; indicate the number of PAH and number of rings considered in this summation to provide context for these numbers.

9. Likewise, the phrase "total PAH" must not be used in describing the measurements from this study, as not all PAH isomers were quantified. Instead "measured PAH" should be used throughout, e.g. in the caption for Table SI-1.

10. The following clarifications to the methods are needed: a) why is hexamethylbenzene used as an internal standard? A number of PAH internal standards are reported in the SI; what is the relationship to this compound? b) do ambient measurements correspond to local / ambient temperature and pressure, or standard conditions?; c) GC film thickness is needed on page 7 line 12; d) number of field and laboratory blanks analyzed; e) detection of analytes (other than naphthalene) in field blanks; f) treatment of field blanks, e.g. field blank subtraction; g) number of spike samples; h) preparation of spike samples; i) a section describing statistical analysis software and methods (e.g. ANOVA).

11. The statement about "similar altitudinal distributions" on page 11, line 4 needs clarification. Does this refer to similar logarithmic distributions in prior studies? References are needed.

12. A value and corresponding reference is needed on page 16 line 1 for the value for wheat burning.

13. In SI-1, revise to read "70 eV" and "-20 C until injection."
* * *

---

## Author Comment (AC1) · 27 Jun 2016

We are grateful to the reviewer's thoughtful and illuminating comments and have now amended the manuscript according to their points. We have acknowledged the valuable contribution made by the reviewers in this manuscript. A detailed response to each of the reviewer's points is provided below and we have carefully revised the manuscript (all revisions are highlighted in the text).

5

Reviewers comments 1:

This is an interesting manuscript dealing with the measurement of particulate PAH concentrations in the Himalayan region across two transects going form Nepal into China Tibetan Plateau. The manuscript has new collected data that permits further insights into the background contamination of Himalayan

10   region and the long distance transport of particulate PAH pollution form the Indian subcontinent.

In my opinion (from a non-born English speaker), the quality of the English in the paper needs to be improved. In the text there are some sections that also need improvement.

Answer: The manuscript has been edited by one professional editor (Dr. Dave Chandler; www.GeoEditing.co.uk) who is native English speaker. All changes according to reviewers comments

15   are marked in blue in the text. And some sections also have rewritten to make them logical and clear.

Section 3.1 about the spatial distribution and seasonal trend of TSP and PAH concentrations is presented in a confusing way and some of the declarations about spatial and seasonal trends are not evident when compared with concentration data in Tables and Figures. For example in lines 15-21, page

20   9, there is the statement of a seasonal variation of concentrations which seems to be contradicted by ANOVA tests.

Answer: Have improved this section. The seasonal trends are not evident at remotes sites from the Figure 2. When we calculated the seasonal averaged concentrations of TSP and PAH, the lowest concentrations were observed during the monsoon season. We have added these data in supporting

information (Table SI-1). As for the description in lines 15-21, we calculated again, and the result showed that the seasonal variation of each site was not significant (at the 0.05 level). It was consistent with what we expressed before.

On lines 7-10, page 10, it is concluded that there is an increase in concentrations during the non-monsoon season which, at least for TSP, is not evident from inspection of Figure 2.

Answer: Although the seasonal variation of TSP concentration was not apparent at Dhunche and other remote sites, the calculated results showed that seasonal averaged TSP concentrations during the non-monsoon season (including the pre-monsoon, post-monsoon and winter seasons) were higher than those in the monsoon season. As mentioned above, we added the seasonal averaged TSP and PAH concentrations in the supporting information (Table SI-1).

Section 3.4 concerning the estimation of dry deposition fluxes also needs improvement.

Answer: After careful consideration, we decided to delete this section according to suggestions of the editor and another reviewer.

Why there was a choice of a dry deposition velocity of 1.4 cm/s for calculating the PAH dry deposition fluxes, when there is a so large variability in measured fluxes? Is it an average of bibliographic data?

Answer: As mentioned above, we have deleted this section.

Specific comments

Figure 1. The background site in China it is expressed as Nielamu while in the text the local is referred as Nyalam—correct.

Answer: Have changed Nielamu to Nyalam in Figure 1.

Abstract. IGP is only defined lately in the text. Introduce definition here.

Answer: Have defined the IGP (Indo-Gangetic Plain) in the abstract section.

5    Page 6, line 15-20, define the size cutting characteristics of the TSP cyclone.

Answer:  All suspended particles can be sampled by the TSP cyclone. Have added this information on page 7 line 6 "--with a TSP cyclone which can collect all the suspended particles".

Page 8, line 7, change "bland" to "blank".

10   Answer: Have changed.

Page 9, line 13, change "pokhara" into "Pokhara".

Answer: Have changed.

15   Page 9, line 16, change "variation" to "variations".

Answer: Have changed.

Page 11, line 6, define POPs, HCBs and PCBs.

Answer: Have defined. The POPs, HCBs, and PCBs are persistent organic pollutants (POPs),

20   hexachlorobenzene (HCB) and polychlorinated biphenyls (PCBs), respectively.

Page 12, line 6, change "continents" to "emission sources".

Answer: Have changed.

Page 13, line 14, this is an estimation/calculation and not a measurement.

Answer: Have deleted this section (3.4 Dry deposition fluxes estimation) according to the suggestion of editor and another reviewer.

5 Figure 4, the site Bode with data taken from Chen et al., (2015) is presented in the figure but not discussed in the text. Please discuss.

Answer: Have discussed in the text on page 13 lines 6-8. "However, this is in contrast to Kathmandu where there is a relatively large contribtuion of 4-ring PAHs, reflecting the dominant influence of fossil fuel emissions, for example, vehicle engine exhausts and coal combustion (Chen et al., 2015)."

10

15

20

---

## Author Comment (AC2) · 27 Jun 2016

We are grateful to the reviewer's thoughtful and illuminating comments and have now amended the manuscript according to their points. We have acknowledged the valuable contribution made by the reviewers in this manuscript. A detailed response to each of the reviewer's points is provided below and we have carefully revised the manuscript (all revisions are highlighted in the text).

5 Reviewers comments 2:

This manuscript fail to provide any new information or concept on the sources, occurrence and atmospheric transport of PAHs. This is largely due to very poor quality of discussion throughout the manuscript. There are several hand-waving and qualitative statements in addition to poor concepts. Overall, the manuscript is extremely poor in English (with efforts of 11 authors), making it difficult at

10 several places to understand what is being conveyed. Another irrelevant section in the manuscript pertains to estimation of dry-deposition fluxes of PAHs. It is not clear why it is relevant and important to estimate PAHs fluxes at the six sampling sites. This manuscript cannot be recommended for publication in ACP even after major revision. Authors have presented data for only two parameters, TSP and PAHs, which is not adequate to merit publication in in ACP. I would not like to go through the

15 manuscript again.

General comments:

1) Overall, the manuscript is poor in English, making it difficult at several places to understand what is being conveyed.

Answer: The manuscript has been edited by one professional editor (Dr. Dave Chandler;

20 www.GeoEditing.co.uk) who is native English speaker. All changes according to reviewers comments are marked in blue in the text. And some sections also have rewritten to make them logical and clear.

Section on estimation of dry-deposition fluxes of PAHs is irrelevant. It is not clear why it is important to estimate PAHs fluxes at the six sampling sites. The impact on aquatic systems, if any, or human health

has not been addressed at all. Simply reporting the fluxes (rough estimates – as claimed by authors themselves on page 14, lines 14-15) do not make any sense. With large uncertainty in the deposition velocity of any atmospheric constituent, it is conceptually wrong to state the deposition fluxes of PAHs up to 1st or 2nd decimal units (25.9 ug/m2/d at Pokhara, at Nyalam as 6.74 ug/m2/d and so on – Section 3.4, Pages 13-14). The entire concept of using deposition velocity and deposition fluxes is not valid at all for sampling sites located in high altitude regions. The concept may still be applicable for deposition over high altitude lakes.

Answer: Have deleted this section according to the suggestions of the reviewer and editor.

2) Abstract is still very poorly written: Page 2, lines 1-2: 1) Why it is important to understand transport of PAHs across Himalayas? 2) Why only from the Indo-Gangetic Plain?

Answer: As mentioned above, the manuscript has been improved carefully by one professional language editor. PAHs have received considerable attention for their persistence and toxicity, related to carcinogenic and/or mutagenic properties (Bhargava et al., 2004). The Tibetan Plateau is in the immediate vicinity to South Asia, which is one of the most important source region for particle pollutants in the world (Bond et al., 2007; Zhang and Tao, 2009). Pollutants emitted from South Asia can gather in the foothills of the Himalayas and can be lifted to high altitudes through some big valleys (Qiu, 2013) or even travel over the high elevated Himalayas and invade into Tibetan Plateau which are critical to both the South Asian summer monsoon and hydro-glaciological resource variability in the Himalayas (Bonasoni, 2008; Lüthi et al., 2015; Xia et al., 2011). However, the exact knowledge on source regions, transport pathways and time trends of PAHs to the Himalayas remains uncertain due to lack of observed data. Thus, it is important to understand the transport of PAHs across Himalayas. For the studied region, the dominant surface wind direction is from the south during the monsoon season. While in other seasons, northwesterly winds prevail due to the effect of the westerly winds. Most of

these air masses pass through the Indo-Gangetic Plain (IGP). Thus, the transport of pollutants from the IGP is considered to be the most important pathway controlling the levels of PAHs over the Himalayas (Gong et al., 2014; Bonasoni et al., 2010). Thus we focus on in situ PAHs observation across the Himalayas transported from the IGP. We have added the significance of studying PAHs transported from IGP across the Himalayas in the abstract section. " The Himalayas is in the risks of long-range transported atmospheric pollutants (e.g. polycyclic aromatic hydrocarbons (PAHs)). However, knowledge of PAH concentrations, sources and transport pathways remains limited in this region."

References:

Bhargava, A., Khanna, R., Bhargava, S., Kumar, S.: Exposure risk to carcinogenic PAHs in indoor-air during biomass combustion whilst cooking in rural India, Atmos. Environ., 38, 4761-4767, doi: 10.1016/j.atmosenv.2004.05.012, 2004.

Bonasoni, P.L., P; Angelini, F ; Arduini, J ; Bonafe, U ; Calzolari, F ; Cristofanelli, P; Decesari, S ; Facchini, MC ; Fuzzi, S 2008. The ABC-Pyramid Atmospheric Research Observatory in Himalaya for aerosol, ozone and halocarbon measurements. Science of the total environment 391, 252-261.

Bond, T., Bhardwaj, E., Dong, R., R, J., 2007. Historical emissions of black and organic carbon aerosol from energy‐related combustion, 1850–2000. Global Biogeochemical Cycles 21, GB2018.

Gong, P., Wang, X., Yao, T.: Ambient distribution of particulate-and gas-phase n-alkanes and polycyclic aromatic hydrocarbons in the Tibetan Plateau, Environ. Earth. Sci., 64, 1703-1711, doi: 10.1007/s12665-011-0974-3, 2011.

Lüthi, Z.L., Skerlak, B., Kim, S.W., Lauer, A., Mues, A., Rupakheti, M., Kang, S.C., 2015. Atmospheric brown clouds reach the Tibetan Plateau by crossing the Himalayas. Atmos. Chem. Phys 15, 6007-6021.

Qiu, J., 2013. Pollutants Capture the High Ground in the Himalayas. Science 340, 1042-1042.

Xia, X., Zong, X., Cong, Z., Chen, H., Kang, S., Wang, P., 2011. Baseline continental aerosol over the

central Tibetan plateau and a case study of aerosol transport from South Asia. Atmospheric Environment 45, 7370-7378.

Zhang, Y., Tao, S.: Global atmospheric emission inventory of polycyclic aromatic hydrocarbons (PAHs) for 2004, Atmos. Environ., 43, 812-819, doi: 10.1016/j.atmosenv.2008.10.050, 2009.

3) Page 2, lines 20-24: Isomer ratios are expected to help in identifying the specific source signature, whereas, authors have stated that – quote "isomer ratios suggested that atmospheric PAHs from the Nepal sites were mainly associated with emission of biomass, coal burning and petroleum burning". This is a very qualitative statement.

10    Answer: Although with uncertainty, isomer ratios have been widely used to detect combustion-derived PAHs (Tobiszewski and Namiesnik, 2012; Yunker et al., 2002). They can help to identify the specific sources. However, it cannot quantify the contributions from different sources. Therefore, we changed this sentence to "Both IndP/(IndP+BghiP) and Fla/(Fla+Pyr) ratios suggested that atmospheric PAHs from urban and rural sites were mainly associated with emission from biomass burning, coal burning

15    and petroleum combustion. However, the contribution of biomass burning increased at remote sites." to make it clear.

Reference:

Tobiszewski, M., Namiesnik, J.: PAH diagostic ratios for the identification of pollution emission sources, Environ. Pollut., 162, 110-119, doi:10.1016/j.envpol.2011.10.025, 2012.

20    Yunker, M., Macdonald, R., Vingarzan, R., Mitchell, R., Goyette, D., Sylvestre, S.: PAHs in the Fraser River basin: a critical appraisal of PAH ratios as indicators of PAH source and composition, Org. Geochem., 33, 489-515, doi: 10.1016/S0146-6380(02)00002-5, 2002.

4) Conclusion: Page 18, lines 10-15: If inferences are drawn only from AMBTs, then it is not of much

relevance to measure chemical constituents (example PAHs). Their long-range transport cannot be considered "conservative".

Answer: In our study, we measured PAH concentrations and compositions at six sites. The results showed that three remote sites could be affected by pollutant emitted by the IGP. We got the results from the change of PAH concentration and composition along two south-north transects. Not only from air mass backward trajectory (AMBTs). But the Hysplit model was used to support our conclusion here. Atmospheric residence time with respect to airborne transport is related to the behavior of the carrier particles. Most particulate phase PAHs are adsorbed onto particles in the accumulation mode (0.2-2 μm) (Miguel and Friedlander, 1978). These particles only deposit slowly from the atmosphere and, depending on atmospheric conditions, may be airborne for days or even weeks, being transported over long distances (in excess of 1000 km) (Harrison et al., 1996). Thus, PAHs can undergo long-range atmospheric transport and have been identified in the world's most remote parts (Ding et al., 2007; Masclet et al., 1988).

References:

Ding, X., Wang, X.M., Xie, Z.Q., Xiang, C.H., Mai, B.X., Sun, L.G., Zheng, M., Sheng, G.Y., Fu, J.M., Poschl, U.: Atmospheric polycyclic aromatic hydrocarbons observed over the North Pacific Ocean and the Arctic area: spatial distribution and source identification. Atmos. Environ. 2007, 41, 2061-2072.

Harrison, R.M., Smith, D.J.T., Luhana, L.: Source apportionment of atmospheric polycyclic aromatic hydrocarbons collected from an urban location in Birmingham, UK. Environ. Sci & Technol. 30, 825-832.

Masclet, P., Pistikopoulos, P., Beyne, S., Mouvier, G.: Long-range transport and gas particle distribution of polycyclic aromatic hydrocarbons at a remote site in the Mediterranean-Sea. Atmos. Environ. 1988, 22, 639-650.

Miguel, A.H., Friedlander, S.K.: Distribution of benzo(a)pyrene and coronene with respect to particle

size in Pasadena aerosols in the sub-micron range. Atmos. Environ. 1978, 12, 2407-2413.

5) Page 18, line 8: What authors mean by "higher deposition efficiency"? How this is built in the concept of using deposition velocity for PAHs? How deposition efficiency is assessed from the data

5   presented in the manuscript.

Answer: Concentration of high molecular weight PAH decreased from the south to north along two transects. Besides the emission source effect, the deposition efficiency is another reason. PAHs (3+4 ring) and PAH (5+6 ring) behave differently due to their chemical characteristics and nonvolatile PAHs are more efficiently removed from the atmosphere than semi-volatile PAHs. Thus we considered that

10  high molecular weight PAHs have higher deposit velocity than light molecular weight PAH (Schauer et al., 2003). However, we cannot get this from the deposition velocity. Thus we changed the description of this sentence to "while the increase of 3-ring PAHs at the other three remote sites might reflect the gas/particle phase transition or higher deposition efficiency of high molecular weight PAHs during long-range transport."

15  Reference:

Schauer, C., Niessner, R., Poschl, U.: Polycyclic aromatic hydrocarbons in urban air particulate matter: decadal and seasonal trends, chemical degradation, and sampling artifacts. Environ. Sci. Technol., 37, 2861-2868. DOI: 10.1021/es034059s

20  6) Specific Comments:

a) Abstract, Page 2, line 11: What is the concept of logarithmic decreasing pattern of PAHs with increasing elevation? Is this is an empirical relation only applicable to PAHs?

Answer:  The low-elevation sites had higher PAH concentrations, as they are more strongly affected by the local pollutant sources. While the long-range transport of pollutants predominates at the highelevation sites, dry and wet deposition during the long-range transport is likely to be the main reason for the decrease in TSP and PAH concentrations. From our manuscript, both PAH and TSP concentrations have a decreasing pattern with increasing elevation. We assume this relation also exist in other particle related proxies. However, we are not sure it due to lack of measured data. We have deleted this sentence from the abstract.

b) Page 3, lines 14-15: Why study of PAHs in remote sites is needed for the understanding of the atmospheric mechanisms involved in the long-range transport of these pollutants? Which "atmospheric mechanisms" authors are referring to during long-range transport?

Answer: PAHs are persistent in the atmosphere thus can be transported over long-distance to even remote areas such as the Himalayas. The atmospheric mechanisms mean PAH transport and deposit processes. However, the dry deposition fluxes estimation is not valid according to the suggestion. Thus, we removed 3.4 section and this sentence.

c) Page 4, line 8: ABC is not "Asian Brown Cloud". It refers to "Atmospheric Brown Cloud".
Answer: Have changed.

d) Page 5, lines 5-19: Why these sources are not important for the contribution of PAHs measured at the six sampling sites?

Answer: Actually, these sources are very important for the contribution of PAHs at the six sampling sites. We listed the nearby emission sources around the sampling sites here and in section 3.1, we discussed the influence of these sources on the spatial and seasonal distribution of measured PAHs.

e) Page 9, lines 4-8: Based on TSP and PAHs concentrations, it is conceptually incorrect to conclude

impact and transport of pollutants from the IGP in the winter. What about contribution from intermediate/downwind sources.

Answer: Here, we compared our results (Jomsom, TSP: $96.2 \pm 40.8$ μg/m$^3$; PAHs: $11.1 \pm 2.97$ ng/m$^3$) with Barapani (PAHs: 14.1 ng/m$^3$) which measured OC, EC, WSOC, and PAHs of PM$_{2.5}$ in Barapani and concluded that the study region was influenced by the long-range transport of aerosols from the IGP in winter (Rajput et al., 2013). Considering that Barapani is located in the foothill of the Himalayas, which has similar geographical conditions with Jomsom, we inferred that Jomsom might also be affected by emission from the IGP. The intermediate regions can also affect the studied region. However, emissions from these regions are less compared to those of the IGP, thus, we considered that the study is mainly affected by the IGP.

Reference:

Rajput, P., Sarin M., Kundu, S.S.: Atmospheric particulate matter (PM$_{2.5}$), EC, OC, WSOC and PAHs from NE-Himalaya: abundances and chemical characteristics, Atmos. Pollut. Res., 4, 214-221, doi: 10.5094/APR.2013.022, 2013.

f) Page 9, lines 3-5: Std. deviation on PAHs cannot be stated as 5.65 ng/m3, 2.97 ng/m$^3$ and so on. Are these significant up to 2nd decimal units.

Answer: We have changed these significant to 1 decimal unit.

g) Page 10, lines 10-14: It is not clear what authors are trying to infer and convey. It is rather poor discussion on spatial and temporal variability of TSP and PAHs along south-north transects.

Answer: The seasonal variation of PAH and TSP concentrations are clear at urban and rural sites while not apparent at remote sites. However, the seasonal averaged concentrations of TSP and PAH in Zhongba and Nyalam (located in the Tibet-Himalayas) showed similar seasonal variation with those in

Nepal (Table SI-1). Have rewritten this sentence to "Despite decreasing markedly from south to north along these two transects, the TSP and PAH concentrations in Zhongba and Nyalam exhibited similar seasonal variations with those in Nepal, suggesting that the northern side of the Himalayas might have similar sources for atmosphric PAHs".

h) Page 11, lines 7-8: Concentrations of PAHs in soils and variability with altitude is out of context and irrelevant. PAHs in soils cannot be assumed to be derived from atmospheric deposition.

Answer: Have deleted this sentence.

10   i) Page 11, lines 9-10: Which "nearby contaminant sources" authors are referring to?

Answer: Have changed "nearby pollutant sources" to "local pollutant sources" and clarified that in the section.The local pollutant sources: for example the burning of large amounts of crop straw and wood at Lumbini; eleven cement factories and more than fifty other industries along the nearby Lumbini–Bhairahawa industrial corridor; increasing numbers of vehicles and hotels at Pokhara..

15

j) Page 11, line 12: "Thus, less local anthropogenic emissions —— ". What is "less"?

Answer: Those remote sites have few anthropogenic emission sources compared to urban and rural sites. Have changed this sentence to "Local emission of PAHs was very limited and the long-range transport of pollutants from lowland source regions might predominate at the high-elevation sites (e.g. Jomsom

20   and Zhongba). Dramatic decrease in TSP and PAH concentrations may be due to pollutants depletion during the long-range transport."

k) Page 11, lines 12-15: The entire discussion is very qualitative and poorly written.

Answer: As mentioned above, we have rewritten this sentence to "Local emission of PAHs was very

limited and the long-range transport of pollutants from lowland source regions might predominate at the high-elevation sites (e.g. Jomsom and Zhongba). Dramatic decrease in TSP and PAH concentrations may be due to pollutants depletion during the long-range transport."

l) Page 11, line 15: "–thus we just gave a rough estimate of the regression analysis in this study". What is the relevance of giving "rough estimate"?

Answer: It is not precise and we have deleted this sentence.

m) Page 12, line 8:"——- indicating that biomass combustion is the main source for particulate PAHs in Lumbini". How biomass combustion source can be inferred from particulate PAHs? What are the concentrations and ratio of OC and EC?

Answer: Lumbini is a typical rural site based on rice and wheat planting. Biomass such as agricultural wastes, animal dung, and wood, are burned almost throughout the year mainly for cooking activities. Burning of large amounts of agro-residue also occurs in the IGP, especially in the pre- and post-monsoon seasons (Ram and Sarin, 2010). While some brick kilns are mainly operated from January to April (Chen et al., 2015). In addition, the profile of PAHs we observed in Lumbini was similar to that of residue combustion aerosols collected at North India (Rajput et al., 2011). Thus we conclude that biomass burning is the main source of PAHs in atmosphere of Lumbini. The average concentrations of OC and EC at Lumbini were found to be $32.06\pm23.77$ $\mu g/m^3$ and $6.44\pm3.64$ $\mu g/m^3$, respectively. And the average OC/EC ratios in Lumbini were 4.66, 4.31, 5.24, and 6.47 for the pre-monsoon, monsoon, post-monsoon, and winter seasons, respectively (annual average: $4.82\pm2.27$).

Reference:

Rajput, P., Sarin, M., Rengarajan, R., Singh, D.: Atmospheric polycyclic aromatic hydrocarbons (PAHs) from post-harvest biomass burning emissions in the Indo-Gangetic Plain: Isomer ratios and temporal

trends, Atmos. Environ., 45, 6732-6740, doi: 10.1016/j.atmosenv.2011.08.018, 2011.

Ram, K., Sarin, M.: Spatio-temporal variability in atmospheric abundances of EC, OC and WSOC over Northern India, J. Aerosol. Sci., 41, 88-98, doi: 10.1016/j.jaerosci.2009.11.004, 2010.

Chen, P. F., S. C. Kang, C. L. Li, M. Rupakheti, F. P. Yan, Q. L. Li, Z. M. Ji, Q. G. Zhang, W. Luo and M. Sillanpaa, 2015. Characteristics and sources of polycyclic aromatic hydrocarbons in atmospheric aerosols in the Kathmandu Valley, Nepal. Science of the Total Environment 538, 86-92.

n) Page 15, lines 16-19: There is no new understanding emerging from this qualitative discussion.

Answer: As mentioned in point 3, isomer ratios cannot quantify the contributions from different sources. However, they have been proved helpful to detect combustion-derived PAHs. From the data, we can found that these ratios are different from those in Lumbini and Jomsom, reflecting influence of local emissions at Lumbini. However, the urbanization development of Pokhara is far less than Kathmandu; thus, pollutants from local sources can affect Pokhara itself but not its adjacent region such as Jomsom.

o) Page 17, lines 1-2: What is "thermally driven flows through Himalayan valleys and up sides?

Answer: Have changed this sentence "--thermally driven flows through Himalayan valleys."

p) Page 17, lines 3-4: "—— diurnal valley wind system often occurs that blows up valley–". What authors mean by "that blows up valley—- "?

Answer: Have changed this sentence to "A diurnal valley wind system often occurs in mountainous terrain, with an up-valley wind during the day reversing to a down-valley wind during the night."

7) There are several confusing & qualitative statements:

Page 2, lines 22-24;

Answer: Have changed this sentence to "Similar compositions were found at three remote sites located on both sides of the Himalayas (Jomsom, Zhongba, and Nyalam), suggesting that the northern side of the Himalayas may be affected by anthropogenic emissions from the IGP via long-range atmospheric transport."

Page 3, lines 14-15;

Answer: As mentioned in specific comments of (b), we deleted this sentence.

Page 9, lines 7-10;

10 Answer: Have changed this sentence to "Considering that local emissions in Jomsom are very low, the measured pollutants probably accumulate by the up-valley winds that transport pollutants from the IGP."

Page 10, lines 10-15;

15 Answer: Have changed this sentence to "Despite decreasing markedly from south to north along these two transects, the TSP and PAH concentrations in Zhongba and Nyalam exhibited similar seasonal variations to those in Nepal, suggesting that the northern side of the Himalayas might have similar sources for atmospheric PAHs."

20 Page 11, lines 6-7; 10-15;

Answer: Have changed this sentence to "For example, Gong et al. (2014) observed that the concentrations of persistent organic pollutants (POPs) such as hexachlorobenzene (HCB) and polychlorinated biphenyls (PCBs) in the atmosphere decreased by a factor of approximately 3 times from 135 m to 5100 m." and "Local emission of PAHs was very limited and the long-range transport of

16

pollutants from lowland source regions might predominate at the high-elevation sites (e.g. Jomsom and Zhongba). Dramatic decrease in TSP and PAH concentrations may be due to pollutants depletion during the long-range transport", respectively.

Page 13, lines 3-4;

Answer: Have changed this sentence to "In contrast, the PAH profiles in Zhongba and Nyalam were similar to that in Jomsom (Fig. 4), which might also reflecting the long-range transport of pollutants from south to north across the Himalayas".

Page 14, lines 12-15;

Answer: As mentioned in the point 1, we deleted this section according to the reviewer's and editor's suggestion.

Page 15, lines 16-19

Answer: Have changed the sentence to "a significant level of rural biomass burning ocurrs around the city. Thus, the PAHs in Pokhara were derived from the combined impacts of local contributions from Pokhara city (both biomass and fossil burning) and polluted air mass transport from upwind IGP which often arrived during the winter and pre-monsoon seasons."

8) English errors:

Abstract, Page 2, line 11: "exhibited"; wrong English;

Answer: Have changed "exhibited" to "showed".

Page 3, line 1: "long-range transportation"

Answer: Have changed "long-range transportation" to "long-range atmospheric transport".

Page 6, line 17: "pre-burned"; What is pre-burned? It should be "pre-combusted"!
Answer: Have changed "pre-burned" to "pre-combusted".

Page 8, line 2: "All analytic"; What is analytic?
Answer: Have changed "analytic" to "analytical".

Page 10, lines 14 and 20: "long-range transported pollution"? Wrong English "transported"!

10    Answer: Have changed "long-range transported pollution" to "long-range transport pollution".

Page 10, line 21: "—- concentrations "exhibited"—-.
Answer: Have changed "exhibited" to "showed".

15    Page 11, line 9: "The low elevation sites displayed—-"; displayed is not a correct word to use!
Answer: Have changed "displayed" to "showed".

Page 12, line 15: "PAHs are present"; not is present.
Answer: Have changed "PAHs is present" to "PAHs are present".

20

Page 12, line 16: "— PAHs scarcely reenter the atmosphere—-". Very poor English!
Answer: Have changed this sentence to "PAHs were rarely returned to the atmosphere".
Page 14, line 21: "—– indicates non-burned petroleum—–". Non-burned is incorrect word.
Answer: Have changed "non-burned" to "non-combusted".

18

---

## Author Comment (AC3) · 27 Jun 2016

We are grateful to the editor's thoughtful and illuminating comments and have now amended the manuscript according to the points. A detailed response to each of the reviewer's points is provided below and we have carefully revised the manuscript (all revisions are highlighted in the text). And we also improved the manuscript with the help of Dr. Dave Chandler (www.GeoEditing.co.uk).

5    Editor comments:

In addition to the comments made by referees, please address the following comments and suggestions:

1. The review of prior studies of PAH in the Himalayas on page 4 line 15-16 is very terse and has overlooked several relevant publications that provide insight to PAH sources and seasonal variation in the Himalaya:

10   Chen, P. F., S. C. Kang, C. L. Li, M. Rupakheti, F. P. Yan, Q. L. Li, Z. M. Ji, Q. G. Zhang, W. Luo and M. Sillanpaa, 2015. Characteristics and sources of polycyclic aromatic hydrocarbons in atmospheric aerosols in the Kathmandu Valley, Nepal. Science of the Total Environment 538, 86-92.

Kim, B. M., J. S. Park, S. W. Kim, H. Kim, H. Jeon, C. Cho, J. H. Kim, S. Hong, M. Rupakheti,

A. K. Panday, R. J. Park, J. Hong and S. C. Yoon, 2015. Source apportionment of PM10 mass and

15   particulate carbon in the Kathmandu Valley, Nepal. Atmospheric Environment 123, 190-199.

Stone, E. A., J. J. Schauer, B. B. Pradhan, P. M. Dangol, G. Habib, C. Venkataraman and V. Ramanathan, 2010. Characterization of emissions from South Asian biofuels and application to source apportionment of carbonaceous aerosol in the Himalayas. Journal of Geophysical Research Atmospheres 115.

20   Answer: Have discussed these references in detail. "Some studies of elements and carbonaceous particle concentrations have already been carried out in this region. For example, Bonasoni et al. (2010) provided a detailed description of the atmospheric conditions in the high Himalayas, revealing that brown cloud hot spots mainly influence the South Himalayas during the pre-monsoon season when BC and $PM_1$ values are higher in comparison to other seasons; Kim et al. (2015) identified and quantified

19

the contributions of different sources (including brick kilns, motor vehicles, fugitive soil dust and biomass/refuse burning) to particulate carbon in the Kathmandu Valley, using the Solver for Mixture Problem receptor model; Stone et al.(2010) characterized the emission profiles of various biofuels from South Asia and applied them to Godavari, located on the southern edge of the Kathmandu Valley, to improve source apportionment of carbonaceous aerosol in South Asia."

2. In light of the abovementioned source apportionment studies – there is evidence for multiple sources of PAH (e.g. coal, biomass, and fossil fuel use) in the Himalayas. The limitations of using PAH isomer ratios for source identification in the presence of multiple sources should be discussed.

Answer: Have discussed the limitation of using PAH isomer ratios for source identification in section 3.4. "It should be noted that while ratios can be somewhat helpful in distinguishing petrogenic from combustion-derived sources, the diversity of fuels and combustion conditions is likely to produce variations in ratios from a single source, hindering the identification of biomass versus fossil fuel combustion inputs. Additionally, PAHs can be transformed by atmospehric processes so that diagnostic ratios measured in atmospehric samples can differ greatly from those reported for the original sources. As a result, source diagnostic ratios should be used with care and in the context of the studied area."

3. Isomer ratios of PAH have also been utilized as a measure of atmospheric aging, particular photochemical degradation (see Bi et al. 2003). Use of the appropriate isomer ratios to track aging may be useful in establishing quantitative support to evaluate local versus long range transport of PAH.

Bi, X. H., G. Y. Sheng, P. Peng, Y. J. Chen, Z. Q. Zhang and J. M. Fu, 2003. Distribution of particulate- and vapor-phase n-alkanes and polycyclic aromatic hydrocarbons in urban atmosphere of Guangzhou, China. Atmospheric Environment 37 (2), 289-298.

Answer: This paper used BeP/BeP+BaP ratio to evaluate local *versus* long range transport of PAHs. Unfortunately, we didn't detect BeP in this study thus could not use this ratio for the particle aging.

4. I concur with referees concerns with the validity of dry deposition flux estimation. A "rough estimation" using an assumed, untested, and unexplained deposition velocity is not valid. The only conclusion drawn is that flux trends follow concentration, which is obvious from Equation 1 when assuming a fixed deposition velocity. The resulting data are not used "assess the atmospheric environment and its impacts on the Himalayan ecosystem" as noted on page 14 lines 14-15. Consequently, section 3.4 should be removed from the manuscript.

Answer: We agree with the suggestion. In the revision, we deleted this section.

5. In the abstract, a number of improvements are needed: a) clarify the importance of studying PAH in the Himalayas ("understanding: : : remains limited" is too vague);

Answer: Have added this sentence in the abstract "The Himalayas is in the risks of long-range transported atmospheric pollutants (e.g. polycyclic aromatic hydrocarbons (PAHs)). However, knowledge of PAH concentrations, sources and transport pathways remains limited in this region."

b) the names of the sites (with their altitudes) should be listed following "Himalayas:" at line 5;

Answer: The name and their altitudes have been listed (Lumbini: 100 m a.s.l; Pokhara: 813 m a.s.l; Jomsom: 3048 m a.s.l; Zhongba: 4704 m a.s.l; Dhunche: 2051 m a.s.l; Nyalam: 4166 m a.s.l).

c) define x and y in the equations at lines 12-13.

Answer: From our manuscript, both PAH and TSP concentrations have a decreasing pattern with increasing elevation. However, The low-elevation sites had higher PAH concentrations, as they are

21

more strongly affected by the local pollutant sources. While at the high-elevation sites (e.g. Jomsom and Zhongba), dramatic decrease in TSP and PAH concentrations may be due to pollutants depletion during the long-range transport. We assume this relation also exist in other particle related proxies. But we are not sure it due to lack of measured data. Thus we deleted this sentence from the abstract.

6. In the introduction (page 3 line 15) clarify what "atmospheric mechanisms" specifically need to be understood and why.

Answer: The atmospheric mechanisms mean PAH transport and deposit processes. However, the dry deposition fluxes estimation is not valid according to the suggestion. Thus, we removed 3.4 section and

10    this sentence.

7. The motivation to study PAH should be justified and clarified in the introduction. PAH generally have low acute toxicity to humans, and their most significant endpoint is cancer.

Answer: Have clarified the motivation to study PAH in the introduction. "PAHs have received

15    considerable attention owing to their persistence and toxicity, especially their carcinogenic and/or mutagenic properties (Bhargava et al., 2004). The World Health Organisation (WHO) recommends guidelines in terms of a carcinogenic slope factor, and the European Union indicative limit value is set at 1 ng/m$^3$ of benzo[a]pyrene (WHO, 2006; European Union, 2005). The United Kingdom has set an air quality standard of 0.25 ng/m$^3$ benzo[a]pyrene (EPAQS, 1999)".

20

8. The "sum of PAH" noted on page 3 line 22 is not operationally defined by the method of analysis; indicate the number of PAH and number of rings considered in this summation to provide context for these numbers.

Answer: Have changed "the sum of PAHs" to "16 PAHs (gas + particulate phase)".

22

9. Likewise, the phrase "total PAH" must not be used in describing the measurements from this study, as not all PAH isomers were quantified. Instead "measured PAH" should be used throughout, e.g. in the caption for Table SI-1.

Answer: Have changed to "measured PAH" throughout this manuscript and the table captions.

10. The following clarifications to the methods are needed:

a) why is hexamethylbenzene used as an internal standard? A number of PAH internal standards are reported in the SI; what is the relationship to this compound?

Answer: We made a mistake here. In our study, 20 ng of anthracene-D10 and benzo(ghi)perylene were added to the eluent solvent as internal standards. We have changed this information in the manuscript on page 18 lines 21-22.

b) Do ambient measurements correspond to local / ambient temperature and pressure, or standard conditions?;

Answer: Yes. The air volume was converted to standard conditions using atmospheric pressure and ambient temperature monitored at each site. We have described this on page 7 lines 16-18.

c) GC film thickness is needed on page 7 line 12;

Answer: The film thickness is 0.25 μm. Have added this information.

d) number of field and laboratory blanks analyzed;

Answer: Have added this information in the manuscript: "Tweleve laboratory and 24 field blanks were extracted and analyzed in the same way as the samples".

23

e) detection of analytes (other than naphthalene) in field blanks;

Answer: Have clarified the detection of analytes in field blanks: "In the field blank samples, the average Aecl, Ace, Phe, and Ant concentrations were detected to be 0.59, 0.45, 0.68, and 0.36 pg/m$^3$, respectively. Other species were below the method detection limit."

f) treatment of field blanks, e.g. field blank subtraction;

Answer: As mentioned above, Tweleve laboratory and 24 field blanks were extracted and analyzed in the same way as the samples. The PAH concentrations were blank corrected but not corrected for the recoveries.

g) number of spike samples;

Answer: Each site, we chose 2 filters as spike samples.

h) preparation of spike samples;

Answer: Each filter was cut to two pieces (Sample$_1$ and Sample$_2$). Sample$_1$ was pretreated in the same way as we described in the manuscript. Add a known quantity (close to the PAH concentrations measured in Sample$_1$) of mixture solution of 16 measuring PAHs to Sample$_2$, then treated that in the same way as sample$_1$. Finally calculate the recovery by using the PAH concentrations of Sample$_1$, Sample$_2$ and added concentration. The recoveries were in the range of 61%-95%. Thus we considered that the pretreat method we used is applicable. We have added this information in the text (Page 8 lines 18-23).

i) a section describing statistical analysis software and methods (e.g. ANOVA).

24

Answer: Have added the section according to the suggestion. "All of the statistical analysis was conducted using SPSS software (Statistical Package for the Social Sciences, Ver.16.0) and included a regression analysis. Comparisons and correlations were considered statistically significant when $p<0.05$. The data are summarized as the mean $\pm$ standard deviation".

11. The statement about "similar altitudinal distributions" on page 11, line 4 needs clarification. Does this refer to similar logarithmic distributions in prior studies? References are needed.

Answer: Have clarified this sentence and added the reference. "Previous studies have reported a clear decreasing pattern with increasing elevation of persistent organic pollutants in remote mountain areas, either in air, soil, or plants (Gong et al., 2014)". The reference just showed a decreasing trend of pollutants.

Gong, P., Wang, X., Li, S., Yu, W., Li, J., Kattel, D., Wang, W., Devkota, L., Yao, T., Joswiak, D.: Atmospheric transport and accumulation of organochlorine compounds on the southern slopes of the Himalayas, Nepal, Environ. Pollut., 192, 44-51, doi: 10.1016/j.envpol.2014.05.015, 2014.

15

12. A value and corresponding reference is needed on page 16 line 1 for the value for wheat burning.

Answer: Have added the value (0.43) and reference (Rajput et al., 2011).

Rajput, P., Sarin, M., Rengarajan, R., Singh, D.: Atmospheric polycyclic aromatic hydrocarbons (PAHs) from post-harvest biomass burning emissions in the Indo-Gangetic Plain: Isomer ratios and temporal trends, Atmos. Environ., 45, 6732-6740, doi: 10.1016/j.atmosenv.2011.08.018, 2011.

20

13. In SI-1, revise to read "70 eV" and "-20 C until injection."

Answer: Have revised.

25